# Episodic Future Thinking Mechanism for Multi-agent Reinforcement Learning

**Dongsu Lee[†] and Minhae Kwon[†*]**
[†]Department of Intelligent Semiconductors
[*]School of Electronic Engineering
Soongsil University, Seoul, South Korea
movementwater@soongsil.ac.kr, minhae@ssu.ac.kr

## Abstract

Understanding cognitive processes in multi-agent interactions is a primary goal in cognitive science. It can guide the direction of artificial intelligence (AI) research toward social decision-making in multi-agent systems, which includes uncertainty from character heterogeneity. In this paper, we introduce *episodic future thinking (EFT) mechanism* for a reinforcement learning (RL) agent, inspired by cognitive processes observed in animals. To enable future thinking functionality, we first develop a *multi-character policy* that captures diverse characters with an ensemble of heterogeneous policies. Here, the *character* of an agent is defined as a different weight combination on reward components, representing distinct behavioral preferences. The future thinking agent collects observation-action trajectories of the target agents and uses the pre-trained multi-character policy to infer their characters. Once the character is inferred, the agent predicts the upcoming actions of target agents and simulates the potential future scenario. This capability allows the agent to adaptively select the optimal action, considering the predicted future scenario in multi-agent interactions. To evaluate the proposed mechanism, we consider the multi-agent autonomous driving scenario with diverse driving traits and multiple particle environments. Simulation results demonstrate that the EFT mechanism with accurate character inference leads to a higher reward than existing multi-agent solutions. We also confirm that the effect of reward improvement remains valid across societies with different levels of character diversity.[2]

## 1 Introduction

Understanding human decision-making in multi-agent interactions is a significant focus in cognitive science. It provides valuable insights into designing interactions among diverse AI agents within multi-agent systems. Research has shown that humans employ counterfactual or future scenario simulation to enhance decision-making [45, 17, 49]. While *counterfactual thinking*, simulating alternative consequences of past events, has been extensively explored in multi-agent RL (MARL) [34, 9, 52, 3], *episodic future thinking* [1, 24], the ability to anticipate future events, remains underexplored in literature despite its importance in handling multi-agent interactions.

Human beings strive to anticipate future situations to prevent costly mistakes. One naive approach to incorporate this ability into AI is through future trajectory prediction using model-based RL [40, 19, 31, 54, 28, 26]. However, this approach is feasible only if the state transition model is known or easily learnable, which is often not the case in multi-agent systems. The complexity arises from

---

[*]Corresponding author: M. Kwon
[2]Project Web: https://sites.google.com/view/eftm-neurips2024.

the interdependence of state transitions on the actions of both the agent and other agents, making learning the state transition model challenging. Additionally, diverse agent characteristics exacerbate this challenge by introducing a wide range of action combinations and subsequent states. Thus, explicitly integrating character inference functionality regarding other agents into AI is more suitable for accurate future state prediction and optimal decision-making.

Our goal is to develop an EFT mechanism for RL agents, enabling them to make adaptive decisions in a society where agents have heterogeneous characteristics. We formalize this task as a Multi-agent Partially Observable Markov Decision Process (MA-POMDP), a framework tailored to address the RL problem wherein multiple agents operate under partial observation [35, 55]. This study defines a character by reflecting the behavioral preferences of RL agents, which come from different weight combinations on reward components. For instance, in a driving scenario, some drivers prioritize safety, while others prioritize speed, leading to heterogeneous policies and behavioral patterns across agents.

Implementing the EFT mechanism requires two functional modules: a multi-character policy and a character inference module. The multi-character policy embeds behavioral patterns corresponding to characters. It allows the agent to observe partial information of the state in continuous space and handles a hybrid action space consisting of discrete and continuous actions. The character inference module leverages the concept of inverse rational control (IRC) [18, 25] to infer target agents' characters by maximizing the log-likelihood of their observation-action trajectories. Combining these modules equips the agent with EFT functionality, enabling proactive behavior under heterogeneous multi-agent interactions.

To activate the EFT mechanism, the agent initially acts as an observer, collecting observation-action trajectories of target agents. Utilizing the character inference module and collected trajectories, the agent infers target agents' characters. With this knowledge and leveraging a multi-character policy, the agent predicts others' actions and simulates future observations with its action fixed as 'no action.' This mental simulation allows the agent to estimate the observation at the time point when all target agents have taken actions, but the agent still needs to (*i.e.*, has yet to). It enables the agent to select the best action corresponding to the estimated future observation. In summary, the EFT mechanism empowers the agent to behave proactively in heterogeneous multi-agent interactions.

**Summary of contributions:**

- We introduce character diversity in a multi-agent system by parameterizing the reward function. We propose to build the multi-character policy and equip the agent with it to infer the character of the target agent (Section 3).

- We introduce the EFT mechanism for social decision-making. The agent infers the characters of other agents using the multi-character policy, predicts their future actions based on the inferred characters, simulates the corresponding future observations and selects foresighted actions. This mechanism enables the agent to consider multi-agent interactions in its decision-making process (Section 4).

- We verify the proposed mechanism by increasing character diversity in society. Extensive experiments confirm that the proposed mechanism enhances group rewards no matter how high a character diversity level exists in society (Section 5).

## 2   Related Works

**Episodic Future Thinking.** Cognitive neuroscience aims to understand how humans use memory in decision-making. Interestingly, the trend of the brain's regional neural activation regarding counterfactual reasoning (*i.e.*, simulating alternative consequences of the last episode) and future thinking (*i.e.*, simulating episodes that may occur in the future) is similar [1]. In [56], the authors study the relationship between future thinking and decision-making and confirm that humans perform future-oriented decision-making. The decision-making abilities, such as strategy formulation, are also significant in scenarios that require multi-agent interactions, *e.g.*, social decision-making.

There are several studies to endow this ability with an AI agent [62, 37, 61, 30]. In [30] and [37], the authors forecast the next state from a macroscopic standpoint without a prediction of each agent's behavior. In [61], the authors predict the behavior of an agent through a deep Bayesian network

considering the dynamics and the previous surrounding environment information. Even though these studies can infer future information, no strategy formulation incorporated with prediction is suggested. Namely, most existing approaches use future predictions as auxiliary information for the optimization process without incorporating these predictions into the policy explicitly. In this study, we propose the ETF mechanism can predict future observations based on the current state and predicted actions of surrounding agents. Consequently, the agent equipped with this mechanism can select a foresighted action corresponding to the anticipated future observation.

**Model-based Reinforcement Learning.** Model-based RL incorporates an explicit module representing system dynamics, contrasting with model-free RL. Within model-based RL, two approaches exist: utilizing a known dynamic model and learning it during training [19, 32, 40]. Using the dynamic model, the model-based RL approaches predict future trajectories, a pivotal step for network optimization [15, 16, 5, 57, 14]. Notably, approaches such as Dreamer [15] and Model-Based Policy Optimization (MBPO) [16, 14] demonstrate the practical application of these predictions. Dreamer optimizes a value function using the return of the predicted future trajectories, and MBPO trains the policy using the predicted future trajectories as augmented data samples. Furthermore, to tackle multi-agent problems, [5] and [57] extend these concepts by integrating a global model or communication block.

While these methods often exhibit outstanding performance, they assume ideal conditions such as a small number of homogeneous agents and full observability. In reality, agents encounter incomplete and noisy data, and accurately modeling system dynamics is challenging due to complex interactions between multiple agents with unique behavioral characteristics. This work addresses a partially observable agent in a multi-agent environment with heterogeneous characteristics across agents. We allow the agent to infer other agents' characters and make decisions based on predictions of upcoming observations.

**Machine Theory of Mind.** Human decision-making in social contexts often involves considering multiple perspectives, including the behavioral characteristics of others. This capacity, known as Theory of Mind (ToM) in cognitive science, primarily involves deducing internal models of others and predicting their future actions [2, 20]. AI research aimed at providing machines with this capability has gained attention for enhancing multi-agent system performance, such as machine ToM [42, 41], inverse learning [43, 18, 33], and Bayesian ToM [60]. These approaches aim to reconstruct the target agent's belief, reward function, or dynamic model based on its trajectories. However, they often operate in simple settings, limiting their applicability to scenarios with a small number of agents, a small discrete action space, or minimal character diversity across agents.

In contrast to previous work, this study explicitly develops a character inference module focusing on establishing a link between trajectories and characters. This module allows the target agent's behavior to be explained by character, aligning with the researcher's interests. Additionally, it extends the action space from continuous to hybrid.

**False Consensus Effect.** Psychological research has identified a cognitive bias in humans to assume that their character, beliefs, and actions are common among the general population [10, 6, 7], termed the False Consensus Effect (FCE) [53, 12, 47]. Recent studies suggest that AI may exhibit this false belief [42]. To underscore the importance of character inference in heterogeneous multi-agent scenarios, we compare the performance of the EFT mechanism with two types of agents: the proposed agent, equipped with the character inference module, and the FCE-based agent, which assumes that target agents share the same character as the agent.

# 3 Character Inference Using Multi-character Policy

We aim to build an agent to make optimal decisions under multi-agent interactions. It requires the agent to be able to anticipate the near future by predicting other agents' actions. The agent should possess the ability to infer the others' characters, leveraging observation of their behaviors.

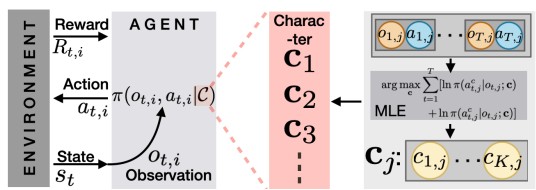

Figure 1: A block diagram of an agent $i$ with a multi-character policy $\pi(o_{t,i}; \mathcal{C})$, where $\mathcal{C}$ is character space. The agent can infer the character $\mathbf{c}$ of others by using the maximum likelihood estimation. Herein, $K$ means the dimension of character vector $\mathbf{c}$.

Accurate character inference is a prerequisite for the EFT mechanism since the character is a crucial clue to predicting future action. Therefore, this section proposes two functional modules for character inference: a multi-character policy and character inference. An illustrative explanation of these functionalities is presented in Figure 1.

## 3.1 Problem Formulations for Multi-agent Decision-making

We consider multi-agent scenarios where RL agents adaptively behave to each other. All agents have to make decisions and execute actions simultaneously, unlike the extensive-form game [36] in which the agents alternate executing the actions.

The multi-agent decision-making problem can be formalized as a MA-POMDP $M = \langle E, \mathcal{S}, \{\mathcal{O}_i\}, \{\mathcal{A}_i\}, \mathcal{T}, \{\Omega_i\}, \{R_i\}, \gamma \rangle_{i \in E}$ that includes an index set of agents $E = \{1, 2, \cdots, N\}$, continuous states $s_t \in \mathcal{S}$, continuous observations $o_{t,i} \in \mathcal{O}_i$, hybrid actions $a_{t,i} = \{a_{t,i}^c, a_{t,i}^d\} \in \mathcal{A}_i$, where continuous action $a_{t,i}^c \in \mathcal{A}_i^c$ and a discrete action $a_{t,i}^d \in \mathcal{A}_i^d = \{w : |w| \leq W, \ w \in \mathbb{Z}, \ W \in \mathbb{N}\}$, where the size of discrete action space is $|\mathcal{A}_i^d| = 2W + 1$, $\mathbb{Z}$ denotes the set of integers, and $\mathbb{N}$ denotes the set of natural numbers. Let $\mathcal{A} := \mathcal{A}_1 \times \mathcal{A}_2 \times \cdots \times \mathcal{A}_N$. Subsequently, $\mathcal{T} : \mathcal{S} \times \mathcal{A} \to \mathcal{S}$ is the state transition probability; $\Omega_i : \mathcal{S} \to \mathcal{O}_i$ is the observation probability; $R_i : \mathcal{S} \times \mathcal{A}_i \times \mathcal{S} \to \mathbb{R}$ denotes the reward function that evaluates the agent's action $a_{t,i}$ for a given state $s_t$ and the outcome state $s_{t+1}$; $\gamma \in [0, 1)$ is the temporal discount factor.

An unordered set of the actions of all agents at time $t$ is denoted as

$$\mathbf{a}_t = \langle a_{t,1}, \cdots, a_{t,i}, \cdots, a_{t,N} \rangle = \langle a_{t,i}, \mathbf{a}_{t,-i} \rangle \in \mathcal{A},$$

where subscript $-i$ represents the indices of all agents in $E$ except $i$. Thus, $\mathbf{a}_{t,-i} = \langle a_{t,1}, \cdots, a_{t,i-1}, a_{t,i+1} \cdots, a_{t,N} \rangle$ represents a set of all agents' actions at time $t$ without $a_{t,i}$. The state transition probability denotes $\mathcal{T}(s_{t+1}|s_t, \mathbf{a}_t)$. Note that state transition is based on the action combination of all agents $\mathbf{a}_t$, not on the action of a single agent $a_{t,i}$.

Next, $\mathbf{c}_i = \{c_{i,1}, c_{i,2}, \cdots, c_{i,K}\} \in \mathcal{C} \in \mathbb{R}^K$ denotes a $K$-dimensional character vector for the agent $i$. Character $\mathbf{c}_i$ can parameterize the reward function of the agent $i$, i.e., $R_{t,i} = R_i(s_t, a_{t,i}, s_{t+1}; \mathbf{c}_i)$. The agent aims to learn the optimal policy that returns the optimal action $a_{t,i}^* \sim \pi^*(\cdot|o_{t,i}; \mathbf{c}_i)$ given observation and character. Specifically, the objective of the agent aims to maximize the expected discounted cumulative reward $\mathcal{J}(\pi) = \mathbb{E}_\pi \left[ \sum_t \gamma^t R_i(s_t, a_{t,i}, s_{t+1}; \mathbf{c}_i) \right]$ by building the best policy $\pi$. This defines the state-action value function $Q^\pi(s, a; \mathbf{c}_i) = \mathbb{E}_\pi \left[ \sum_t \gamma^t R_i(s_t, a_{t,i}, s_{t+1}; \mathbf{c}_i)|s_0 = s, a_0 = a \right]$. In the next section, we discuss the details of the multi-character policy in terms of neural network design and its training.

## 3.2 Training a Multi-character Policy

The multi-character policy includes inputs in continuous space (e.g., observation $o_{t,i}$ and character $\mathbf{c}_i$) and outputs in hybrid space (e.g., action $a_{t,i}$). To build the policy generalized over continuous space, the actor-critic architecture is used. It approximates the policy $\pi_\phi(\cdot|o_{t,i}; \mathbf{c}_i)$ and Q-function $Q_\theta(o_{t,i}, a_{t,i}; \mathbf{c}_i)$, where $\phi$ denotes parameters of the actor network and $\theta$ denotes the parameters of the critic network.

The loss functions used to train the actor and critic networks are $\mathcal{L}(\phi) = -Q_\theta(o_{t,i}, \pi_\phi(\cdot|o_{t,i}; \mathbf{c}_i))$, and $\mathcal{L}(\theta) = |y_t - Q_\theta(o_{t,i}, \pi_\phi(\cdot|o_{t,i}; \mathbf{c}_i))|^2$, respectively. Herein, $y_t = R_{t,i} + Q_{\theta'}(o_{t+1,i}, \pi_{\phi'}(\cdot|o_{t+1,i}; \mathbf{c}_i))$ represents the Temporal Difference (TD) target, where $\theta'$ and $\phi'$ denote the target networks.

Next, we propose a post-processor $g(\cdot)$ to handle hybrid action space. Let a proto-action $\bar{a}_{t,i}^d$ be the output of the actor network. The post-processor $g(\cdot)$ performs quantization process by discretizing the continuous proto-action $\bar{a}_{t,i}^d$ into discrete post-action $a_{t,i}^d$, i.e.,

$$a_{t,i}^d = g(\bar{a}_{t,i}^d, W) = \min \left( \left\lfloor \frac{2W + 1}{2W} \left( \bar{a}_{t,i}^d + \frac{W}{2W + 1} \right) \right\rfloor, W \right), \tag{1}$$

where $\lfloor \cdot \rfloor$ denotes a floor function. The derivation of (1) is presented in Appendix D.

We summarize the multi-character policy training process in Algorithm 1. In the next subsection, we introduce the character inference module that infers the characters of other agents.

### 3.3 Inferring Character of Target Agent

After completing the training on the multi-character policy, our next objective is to infer the character $\mathbf{c}_j$ of the target agent $j \in E$. The agent first collects observation-action trajectories of the target for character inference. Subsequently, it utilizes the multi-character policy to identify the character $\mathbf{c}_j$ that best explains the collected data. To elaborate, $\mathbf{c}_j$ can be estimated by maximizing the log-likelihood of observation-action trajectories $\ln P(o_{1:T,j}, a_{1:T,j}|\mathbf{c}_j)$. This can be formulated as follows.

$$\hat{\mathbf{c}}_j = \arg\max_{\mathbf{c}} \ln P(o_{1:T,j}, a_{1:T,j}|\mathbf{c}) = \arg\max_{\mathbf{c}} \sum_{t=1}^{T} \left[ \ln \pi(a_{t,j}^c|o_{t,j}; \mathbf{c}) + \ln \pi(a_{t,j}^d|o_{t,j}; \mathbf{c}) \right] \quad (2)$$

The derivation of (2) can be found in Appendix E.

To efficiently perform the inference task, we use the gradient ascent method. It runs the iteration by changing $\mathbf{c}$ toward the direction to increase $\mathcal{U}(\mathbf{c}) = \ln \pi(a_{t,j}^c|o_{t,j}; \mathbf{c}) + \ln \pi(a_{t,j}^d|o_{t,j}; \mathbf{c})$, which is summarized in Algorithm 2.[3]

## 4 Foresight Action Selection Based on Episodic Future Thinking Mechanism

This section presents the proposed EFT mechanism that enables the agent to simulate the subsequent observations and to select a foresighted action. The proposed EFT mechanism comprises a future thinking module and an action selection module.

The future thinking module includes two steps: action prediction and the next observation simulation. With these two steps, the agent can foresee the next observation. This process is illustrated in Figure 2. Subsequently, the action selection module enables the agent to decide the current action corresponding to the simulated next observation.

### 4.1 Future Thinking: Step I - Action Prediction

In this step, the agent with the multi-character policy predicts the actions of the neighbor agents by using pre-inferred characters and observations. The agent can predict the action of the target agent $j \ (\in E, j \neq$

---

**Algorithm 1** Multi-character policy training

**Initialization:** Actor network $\phi$, critic network $\theta$
**Require:** Total episode $M$, total time steps per episode $T$, discrete action space $W$, agent $i$
**for** episode $m = 1, M$ **do**
    Reset $s_1$ and get $o_{1,i} \sim \Omega_i(\cdot|s_1)$
    Sample character $\mathbf{c}_i \sim \mathcal{C}$
    **for** timestep $t = 1, T$ **do**
        Get proto-action $\{a_{t,i}^c, \bar{a}_{t,i}^d\} \sim \pi_\phi(\cdot|o_{t,i}; \mathbf{c}_i)$
        Get post-action
        $a_{t,i}^d \leftarrow g(\bar{a}_{t,i}^d, W)$
        Execute $a_{t,i} = \{a_{t,i}^c, a_{t,i}^d\}$, Update $s_{t+1}$
        Receive $R_{t,i}$, Get $o_{t+1,i} \sim \Omega_i(\cdot|s_{t+1})$
        Calculate $\mathcal{L}(\phi), \mathcal{L}(\theta)$, Update $\phi, \theta$
    **end for**
**end for**
**return** $\phi, \theta$

---

**Algorithm 2** Character inference module

**Require:** Trained actor network $\phi$, length of trajectories $T$, trajectories $o_{1:T,j}, \{a_{1:T,j}^c, a_{1:T,j}^d\}$, and initial $\mathbf{c} \sim \mathcal{C}$, target agent $j$
**repeat**
    Reset $\mathcal{U}(\mathbf{c}) = 0$
    **for** $t = 1, T$ **do**
        Calculate $\mathcal{U}(\mathbf{c})$ using Eq. 2
    **end for**
    Update $\mathbf{c} \leftarrow \mathbf{c} + \alpha \nabla_{\mathbf{c}} \mathcal{U}(\mathbf{c})$
**until** $\mathbf{c}$ converges
**return** $\hat{\mathbf{c}}_j \leftarrow \mathbf{c}$

---

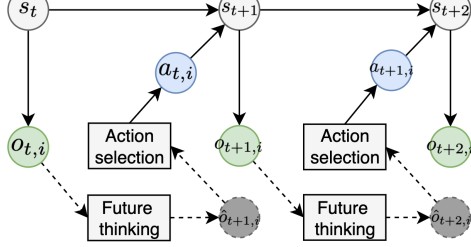

Figure 2: Diagram of POMDP with EFT mechanism. The future thinking and action selection modules are included to obtain action from the observation. The solid lines and circles represent the actual event. The dashed ones depict the virtual event in the simulated world of the agent $i$.

---

[3]By specifying the distribution of $\pi$, (2) can be reformulated. In Appendix F, an example of the Gaussian distribution of continuous action $\pi(a_{t,j}^c|o_{t,j}; \mathbf{c})$ and the Dirac delta distribution of discrete action $\pi(a_{t,j}^d|o_{t,j}; \mathbf{c})$ is provided.

$i)$[4] using the trained multi-character policy $\pi_\phi$ and inferred character $\hat{\mathbf{c}}_j$, *i.e.*, $\hat{a}_{t,j} \sim \pi_\phi(\cdot|o_{t,j}; \hat{\mathbf{c}}_j)$. Therefore, the predicted action set of others $\hat{\mathbf{a}}_{t,-i}$ is as follows.

$$\hat{\mathbf{a}}_{t,-i} = \langle \pi_\phi(o_{t,1}; \hat{\mathbf{c}}_1), \cdots, \pi_\phi(o_{t,i-1}; \hat{\mathbf{c}}_{i-1}), \pi_\phi(o_{t,i+1}; \hat{\mathbf{c}}_{i+1}), \cdots, \pi_\phi(o_{t,N}; \hat{\mathbf{c}}_N) \rangle$$

### 4.2 Future Thinking: Step II - Next Observation Simulation

In this step, we introduce how the agent simulates its next observation by using the predicted action $\hat{\mathbf{a}}_{t,-i}$. Note that this prediction is the result of the mental simulation of agent $i$, when $a_{t,i} = \emptyset$ is satisfied. Herein, $\emptyset$ denotes null action, meaning that no action is performed. This is to simulate the observation of the time point when all target agents performed the action, but the agent has not yet.

The simulated next observation $\hat{o}_{t+1,i}$ can be determined based on the predicted action set $\hat{\mathbf{a}}_{t,-i}$ and the current observation $o_{t,i}$. The function of the next observation simulation $\mathcal{D}(\cdot)$ is defined as $\hat{o}_{t+1,i} = \mathcal{D}(o_{t,i}, \hat{\mathbf{a}}_{t,-i}, a_{t,i} = \emptyset)$. The action selection using the simulated next observation $\hat{o}_{t+1,i}$ allows the agent to ignore the influence of others' actions. This is because the next state is determined solely by its own action $a_{t,i}$ in the agent's mental simulation, as $\hat{o}_{t+1,i}$ has already applied the others' actions $\hat{\mathbf{a}}_{t,-i}$.

### 4.3 Action Selection

Once the agent has simulated the next observation $\hat{o}_{t+1,i}$, the agent can make a fore-sighted decision. The agent uses the multi-character policy $\pi_\phi$ with the input of the simulated next observation $\hat{o}_{t+1,i}$ and its own character $\mathbf{c}_i$, and finally gets the action $a_{t,i} = \{a_{t,i}^c, \bar{a}_{t,i}^d\} = \pi_\phi(\cdot|\hat{o}_{t+1,i}; \mathbf{c}_i)$. In

---

**Algorithm 3** Episodic future thinking mechanism

**Require:** Trained actor-network $\phi$, discrete action space parameter $W$, set of inferred characters $\hat{\mathbf{c}}_{-i}$, character of agent $\mathbf{c}_i$, initial state $s_1$
**for** $t = 1, T$ **do**
 Get observation $o_{t,i} \sim \Omega_i(s_t)$
  // Start future simulation //
 **for** $j = 1, N(j \neq i)$ **do**
  Get observation $o_{1,j} \sim \Omega_j(s_t)$
  Predict action of agents $j$
   $\hat{a}_{t,j} \sim \pi_\phi(\cdot|o_{t,j}; \mathbf{c}_j)$
  Store $\hat{a}_{t,j}$ in predicted action set $\hat{\mathbf{a}}_{t,-i}$
 **end for**
 Simulate future observation of agent $i$
   $\hat{o}_{t+1,i} = \mathcal{D}(o_t, \hat{\mathbf{a}}_{t,-i}, a_{t,i} = \emptyset)$
  // End simulation //
 Get proto-action $\{a_{t,i}^c, \bar{a}_{t,i}^d\} \sim \pi_\phi(\cdot|\hat{o}_{t+1,i}; \mathbf{c}_i)$
 Get post-action $a_{t,i}^d \leftarrow g(\bar{a}_{t,i}^d, W)$
 Execute $a_{t,i} = \{a_{t,i}^c, a_{t,i}^d\}$, Update $s_{t+1}$
**end for**

---

other words, the agent can select an adaptive action with consideration for others' upcoming behaviors. The decision-making procedure with the proposed EFT mechanism is summarized in Algorithm 3.

## 5 Experiments

To select a suitable task that can verify the effectiveness of the proposed solution, we consider the following requirements. There should be multiple approaches to achieving character diversity, as well as interactions between agents. The agent should have only partial observations of the state, and the action space should be both continuous and discrete.

We chose the autonomous driving task, which has numerous automated vehicles on the road. The task can consider the driving character of the agent based on driving preferences (*e.g.*, one agent prioritizes safety, and the other prioritizes speed) [46, 4, 50, 23, 22]. Additionally, it is realistic for a driver to behave under the partial observation of the road state, and the driver makes a decision in a hybrid action space. To implement this task, we use the FLOW framework [58, 21, 8]. The scenario includes multiple automated vehicles on the highway. The number of agents $|E| = 21$, and each agent decides on acceleration and lane change control at a given observation. Here, we express the driving character using weights of three reward terms, *i.e.*, $\mathbf{c}_i = [c_{i,1}, c_{i,2}, c_{i,3}]$.[5] The target agent $j$ is limited to the vehicles located in the observable area.

To confirm the scalability of the proposed solution, we also provide simulation results with a multiple particle environment (MPE) [29] and starcraft multi-agent challenge (SMAC) [48], a popular MARL testbed. All results in this section are averaged results of over 10 independent experiments. The

---

[4]If the agent $i$ cannot observe the entire set of agents, a subset of the agent can be the targets of agent $i$, *i.e.*, $E_{\mathcal{O}_i} \subset E$.
[5]Details regarding the experiments are presented in Appendix G.

markers indicate the average value, and the shaded area represents the confidence interval within one standard deviation.

## 5.1 Performance Evaluation: Character Inference

To make the EFT mechanism more effective, an accurate character inference should be preceded. In this subsection, we investigate the character inference module with two questions.

- How many iterations does it require to achieve an accurate inference (in terms of repetition in Algorithm 2)?
- How long should the agent collect the observation-action trajectories of target agents (in terms of trajectory length $T$ in Algorithm 2)?

In Figure 3, the performance of the character inference module is presented. To ignore the effect of the initial point in convergence, the initial point of the character is randomly selected. More results regarding the initial point are provided in Appendix I.

Figure 3**A** illustrates the convergence of the estimated character to the true one. The inaccuracy of inference is evaluated based on the L1-norm between the estimated character and the true one. Thus, a lower L1-norm implies higher inference accuracy. As the number of iterations increases, the L1-norm quickly decreases to approximately zero, meaning that the estimated value quickly converges to the true one. Specifically, if the number of iterations is set to over 500, high accuracy of the character inference can be achieved.

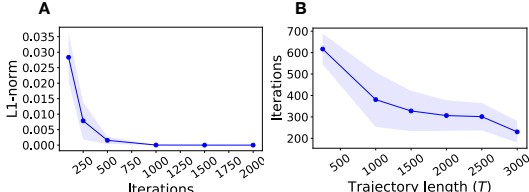

Figure 3: The performance of the character inference module. **A**. L1-norm between estimated and true characters over the number of iterations ($T = 1000$). **B**. The number of required iterations for convergence over the length of the observation-action trajectory $T$.

Figure 3**B** shows the trade-off between the length of observation-action trajectory $T$ and the number of iterations required for the convergence. The convergence criterion is set to L1-norm $\leq 5 \times 10^{-4}$. The results demonstrate that the number of iterations for convergence decreases as longer trajectories are provided. Thus, the length of trajectories and the number of iterations can be jointly determined by considering system requirements.

## 5.2 Ablation Study: Character Inference and EFT Modules

We investigate the impact of two main modules (the character inference module and the EFT module) on performance by increasing character diversity levels of the heterogeneous society. The following three cases are compared.

- `Proposed`: the agent enables the EFT with the inferred character of other agents based on the character inference module.
- `FCE-EFT`: the agent experiences the FCE by assuming that all other agents have equal character to itself (*i.e.*, $\mathbf{c}_j = \mathbf{c}_i, \forall j \in E$). So, no character inference is required. The agent performs the EFT, but action prediction is performed based on the same character $\mathbf{c}_i$.
- `without EFT` (baseline) [11]: the agent performs neither character inference nor the EFT mechanism. It treats the problem as a single agent RL and selects the best action given observation. The policy is trained based on the TD3.

In Figure 4, the average rewards of entire agents are presented over increasing the number of character groups.[6] The higher number of character groups means that more diverse characters coexist in society, and the higher reward implies better performance. Because the number of agents is fixed to $|E| = 21$, the number of members per character group is $|E|/n$, where $n$ denotes the number of groups. The members belonging to the same group have the same character $\mathbf{c}$. Note that a character of each group is randomly sampled from character space $\mathcal{C}$ in every independent experiment.

---

[6]Each market point is the average value of 10 independent test experiments. To obtain all results presented in Figure 4, we run $7 \times 3 \times 10 = 210$ test experiments.

Figure 4 highlights the amount of reward enhancement or degradation by equipping the proposed modules. The proposed approach consistently outperforms the baseline (`without EFT`), and the `FCE-EFT` is inferior to the baseline when character diversity exists. These results verify that the EFT mechanism with accurate character inference always enhances the reward. However, the naive employment of the EFT mechanism with the incorrect character degrades the reward. This is because incorrect character inference leads to incorrect action prediction and next observation simulation, which leads to improper action selection of the agent, leading to low reward. Therefore, accurate character inference is crucial in the EFT mechanism.

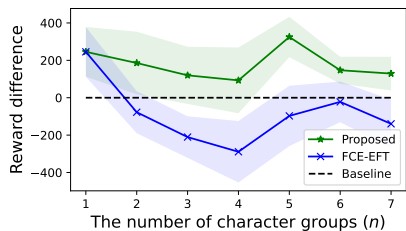

Figure 4: The amount of reward enhancement for two EFT approaches by setting `without EFT` as a baseline (*i.e.*, the reward of other approaches - the reward of `without EFT`).

## 5.3 Investigating the Effects of Trajectory Noise

To infer the character of the target agent, the EFT agent needs to collect observation-action trajectories of the target agent. Since the observations made by the EFT agent towards the target agent may not be perfect (*i.e.*, they could be a noisy version of the target agent's true observations), we further investigate the performance of the proposed EFT framework concerning the accuracy of the collected trajectories. This investigation consists of two steps. First, we look deeply into the effect of trajectory accuracy on character inference, and thereafter, we examine the EFT performance regarding character inference accuracy.

**Character inference with trajectory accuracy.** Table 1 shows the character inference accuracy as the noise level for a collected trajectory increases. As expected, the character inference accuracy decreases as the noise variance increases. Please be aware that the considered standard deviation is not trivial given that our observation range is $[-1, 1]$. Specifically, we provide the signal-to-noise ratio (`SNR`) with a quality level (`Qual`) across each standard deviation. We label the quality of each level based on [13].

Table 1: Character inference accuracy over the standard deviation of trajectory noise. (Accuracy: `ACC`)

|  | Standard deviation of trajectory noise | | | | |
|---|---|---|---|---|---|
|  | 0.01 | 0.05 | 0.1 | 0.2 | 0.3 |
| `ACC` | 99.6 ±0.01 | 98.3 ±0.07 | 91.8 ±0.23 | 81.1 ±0.52 | 69.5 ±0.66 |
| `SNR[dB]` | 34.7 | 21.3 | 14.7 | 9.2 | 4.7 |
| `Qual` | Excellent | Good | Fair | Poor | Poor |

We believe that this result provides valuable insights into the expected performance of our proposed solution, particularly in scenarios where observation prediction technology is deployed.

**EFT performance with character accuracy.** In Figure 5, $x$ and $y$ axes are the accuracy of character inference and average reward, and $n$ is diversity level. As expected, the result shows that the performance of the EFT agent naturally increases when the accuracy of predicted observation increases. Interestingly, the proposed solution holds up the performance even at a char-

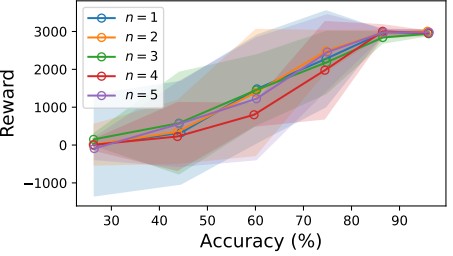

Figure 5: The average reward for increasing the accuracy of character inference.

acter inference accuracy of approximately 90% (*i.e.*, the error rate of 10%). It is also worth mentioning that the performance has a similar trend across the diversity levels, which confirms that the proposed method is robust against diversity levels.

## 5.4 Assessing Generalizability: Inference on Out-of-Distribution Character

It can be impractical and challenging to train all characters within a pre-defined range, and a trained agent can confront an out-of-distribution (OOD) character in the deployment phase. This subsection demonstrates the inference performance on the OOD range of pre-trained agents with specific character samples. To this end, we consider the following two cases:

1. train on $[0.0, 0.6]$ and $[0.8, 1.0]$, thereby inferring on $\{0.65, 0.7, 0.75\}$,
2. train on $[0.2, 0.8]$, thereby inferring on $\{0.0, 0.1, 0.9, 1.0\}$.

Figure 6 shows the average of estimated characters over true ones. The gray dimmed area is the OOD range, which is an unseen character range in the training phase, and red and blue circles present an OOD and in-distribution estimated character value, respectively. Figure 6 **A** represents case 1, where the model appears to perform well in predicting characters in unseen regions. Figure 6 **B** represents case 2, which performs inference on the points outside of the trained range. It is observed that the inference accuracy is slightly declined compared to case 1, but it can still successfully capture the overall pattern by predicting the extreme values that are close to the true ones.

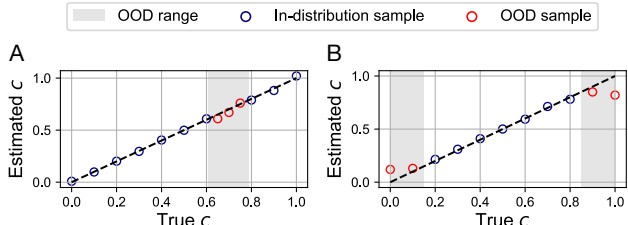

Figure 6: The performance of the character inference module on OOD character range.

## 5.5 Performance Comparisons

We compare the performance of the proposed solution to the following popular MARL, model-based RL, and agent modeling algorithms: MADDPG [29], MAPPO [63], Q-MIX [44], Dreamer [15], MBPO [16], ToMC2 [59], and LIAM [38]. In baseline algorithms, we go through independent policy training regarding the diversity level of society.[7] Note that the proposed method does not need plural training for different heterogeneity settings. See Appendix J for an additional explanation of the baseline algorithms and standard deviation for Table 2.

Table 2 shows the average reward of the entire agents as the number of character groups increases. This result verifies that the proposed solution outperforms all popular MARL algorithms. Note that the MARL algorithms assume centralized training, which requires access to the observations and actions of all agents in policy training. In contrast, our solution trains the policy with only local observations and actions, which can be a more practical solution. The Q-MIX has the lowest performance since it operates in a discrete action space, whereas our task is in a hybrid action space.

Table 2: Performance comparison across diversity level.

| Algorithm | The number of character groups ($n$) | | | | |
| | 1 | 2 | 3 | 4 | 5 |
| --- | --- | --- | --- | --- | --- |
| Proposed | 2899 | **3047** | **2976** | **2948** | **3051** |
| FCE-EFT | 2899 | 2784 | 2646 | 2566 | 2629 |
| MADDPG [29] | 2763 | **3006** | 2800 | **2933** | 2856 |
| MAPPO [63] | 2753 | 2862 | 2597 | 2529 | 2763 |
| Q-MIX [44] | 2199 | 2310 | 2288 | 2118 | 1861 |
| Dreamer [15] | 2911 | 2813 | 2733 | 2631 | 2701 |
| MBPO [16] | 2089 | 1964 | 1523 | 1893 | 1633 |
| ToMC2 [59] | **3016** | 2812 | 2683 | 2691 | 2511 |
| LIAM [38] | 1913 | 1792 | 1771 | 1683 | 1733 |

Table 2 also demonstrates the performance of popular model-based RL algorithms as the diversity level increases. It is obvious that the performance gap between model-based RL and the proposed solution increases as the diversity level increases. In addition, the standard deviation of model-based RL algorithms (provided in Table J1 in Appendix J) is much larger than the proposed solution, which shows the difficulty of learning a dynamic model without understanding others in multi-agent systems. Specifically, Dreamer cannot adapt to high diversity levels, and it has a broader variance than other algorithms. Additionally, the result of MBPO exhibits that it is hard to trust generated transitions from a dynamic model.

In the case of agent-modeling algorithms, ToMC2 achieves the best score in the $n = 1$ scenario, but its performance decreases as the diversity level increases; LIAM fails at all diversity levels. On the other hand, the proposed solution is robust to changes in the surrounding agents and maintains high performance across diversity levels. We conjecture why two baselines fail in this setup, as follows.

---

[7]For each algorithm, five independent trainings are performed since five heterogeneity settings are considered, *i.e.*, $n = [1, 2, 3, 4, 5]$.

ToMC2 requires retraining or adjusting the ToM module as surrounding agents change. The ToM module is tailored to other agents for the prediction of information (*e.g.*, goals, observations, and actions). Next, LIAM also necessitates a new opponent modeling process for each test environment. In addition, prior works on opponent modeling rarely involve more than four players.

## 5.6 Additional Evaluation on MPE and SMAC

Beyond the autonomous driving task, we run the performance comparison on the MPE and SMAC testbed.

**Multiple Particle Environment.** The MPE tasks consider a small number of agents (three or four) and groups (one or two). Therefore, we set the character for each group as a single character, that is, the diversity level $n = 1$. Table 3 shows the performance comparison across each task of MPE. Even though our method is specialized for a high level of character diversity environment, the results demonstrate that the proposed solution is competitive in a simple environment by achieving the best score in two out of three tasks. We provide additional information on the MPE task in Appendix K.

Table 3: Performance comparison with MARL baseline algorithms on MPE tasks. Performance of † denoted algorithm is based on [39].

| Task | MAPPO$^\dagger$ | MADDPG$^\dagger$ | Q-MIX$^\dagger$ | Proposed |
|---|---|---|---|---|
| Spread | $-149.26$ | $-157.10$ | $-154.70$ | $\mathbf{-149.12}$ |
| Adversary | 9.61 | 7.80 | 8.11 | **10.01** |
| Tag | 13.78 | 6.65 | **15.00** | 14.57 |

**StarCraft Multi-agent Challenge.** The setup of SMAC tasks is similar to MPE tasks, *i.e.*, the EFT agent does not need to infer the character because they have the same (character diversity as $n = 1$). Table 4 exhibits the performance on the SMAC tasks. The proposed solution demonstrates superior performance across SMAC tasks, particularly excelling in more complex scenarios like 3s5z_vs_3s6z and 6h_vs_8z. Although MAPPO shows competitive performance, especially in simpler tasks like 2s3z, the proposed method proves more effective overall in handling both simple and complex multi-agent tasks. Additional information in terms of SMAC tasks can be shown in Appendix L.

Table 4: Performance comparison with MARL baseline algorithms on SMAC tasks. Performance of † denoted algorithm is based on [63].

| Task | MAPPO$^\dagger$ | MADDPG | Q-MIX$^\dagger$ | Proposed |
|---|---|---|---|---|
| 2s3z | **100** | 90.3 | 95.3 | 98.8 |
| 3s5z_vs_3s6z | 63.3 | 18.9 | 82.8 | **84.3** |
| 6h_vs_8z | 85.9 | 68.0 | 9.4 | **93.8** |

## 6 Discussion

**Conclusion.** In this paper, we propose the EFT mechanism, which is a social decision-making approach for a multi-agent scenario. The EFT mechanism enables the agent to behave by considering current and near-future observations. To achieve this functionality, we first build a multi-character policy that is generalized over character space. Then, the agent with the multi-character policy can infer others' characters using the observation-action trajectory. Next, the agent predicts the others' behaviors and simulates its future observation based on the proposed EFT mechanism. In the simulation result, we confirm that the proposed solution outperforms existing solutions across all diversity levels of the heterogeneous society.

**Broader Impacts.** The proposed EFT idea paves the way for research on multi-agent scenarios. The proposed method enables the agent to simulate other agents' upcoming actions, which is analogous to humans' decision-making. Furthermore, we believe the proposed method can be broadened by combining counterfactual thinking, current information, and future thinking.

**Limitations.** Even though this work shows promising results with a novel method, there are a few limitations to tackle. In our experiments, there is only one EFT agent, and all other agents do not have the EFT functionality. This is an inevitable setting to make the problem tractable. Additionally, we follow the non-stationary regarding the agent's policy in the training phase and stationary in the execution phase. Since the character is mapped into policy, this stationary property has a connection to the character itself. To improve practicality, we should further investigate how the proposed solution works when the other agent's policy is non-stationary in the execution phase.

## Acknowledgement

This research was supported in part by the National Research Foundation of Korea (NRF) grant (RS-2023-00278812), and in part by the Institute of Information & communications Technology Planning & Evaluation (IITP) grants (No. 2021-0-00739, 2022-2020-0-01602) funded by the Korea government (MSIT). D. Lee is grateful for financial support from Hyundai Motor Chung Mong-Koo Foundation.

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

# Appendix: Episodic Future Thinking Mechanism for Multi-agent Reinforcement Learning

## Contents

# Appendix A    Summary of Notations

| Notation | Description | Notation | Description |
|---|---|---|---|
| $E$ | a set of agents | $i$ | an agent $i$ |
| $\mathcal{S}$ | a state space | $s_t$ | a state at time $t$ |
| $\mathcal{O}_i$ | an observation space of agent $i$ | $o_{t,i}$ | an observation of agent $i$ at time $t$ |
| $\mathcal{A}$ | an action space | $a_{t,i}$ | an action of agent $i$ at time $t$ |
| $a_{t,i}^c$ | a continuous action of agent $i$ at time $t$ | $a_{t,i}^d$ | a discrete action of agent $i$ at time $t$ |
| $\bar{a}_{t,i}$ | a proto-action of agent $i$ at time $t$ | $a_{t,i}^*$ | a true action agent $i$ at time $t$ |
| $\mathbf{a}_t$ | a joint action at time $t$ | $\mathbf{a}_{t,-i}$ | a joint action except agent $i$ at time $t$ |
| $\mathcal{T}$ | a state transition probabilities | $\Omega_i$ | an observation transition probabilities of agent $i$ |
| $R_{t,i}$ | a reward of agent $i$ at time $t$ | $\gamma$ | a temporal discounted factor |
| $\mathcal{C}$ | a character space | $\mathcal{D}$ | a dynamic model |
| $\mathbf{c}_i$ | a character vector of agent $i$ | | |

# Appendix B    System Specification

| CPU | AMD Ryzen 9 3950X 16-core |
|---|---|
| GPU | GeForece RTX 2080 Ti |
| RAM | 128 GB |
| SSD | 1T |

# Appendix C    Hyperparameters

## C.1    Algorithm 1

| Hyperparameter | Value | Hyperparameter | Value |
|---|---|---|---|
| total episodes ($K$) | 3500 | total timesteps ($T$) | 3000 |
| policy delay ($d$) | 2 | target noise variance ($\bar{\sigma}$) | 0.2 |
| replay buffer size ($|\mathcal{B}|$) | $4 \times 10^6$ | train batch size (B) | 128 |
| discount factor ($\gamma$) | 0.99 | soft update rate ($\tau$) | $1 \times 10^{-3}$ |
| exploration variance 1 ($\sigma_1$) | 0.1 | exploration variance 2 ($\sigma_2$) | 0.6 |
| actor learning rate | $5 \times 10^{-4}$ | critic learning rate | $5 \times 10^{-4}$ |
| actor hidden node | $[64, 64]$ | critic hidden node | $[64, 64]$ |
| activation function of actor hidden layer | ReLU | activation function of critic hidden layer | ReLU |
| activation function of actor output layer | tanh | activation function of critic output layer | linear |

## C.2    Algorithm 2

| Hyperparameter | Value | Hyperparameter | Values |
|---|---|---|---|
| optimizer | Adam | learning rate | $10^{-3}$ |
| the number of iterations ($L$) | 200 | the number of samples ($N$) | 3000 |

## Appendix D  Post-processor Function in (1)

To build a post-processor function $g(\cdot)$, we first allocate the continuous action space $\mathcal{A}^d = [-W, W]$ into $|\mathcal{A}^d| = 2W + 1$ discrete action values. In other words, a continuous number lies in the range $\bar{a}_t^d \in \left[ w - \frac{W+w}{2W+1}, w + \frac{W-w}{2W+1} \right]$ is assigned to a discrete action value $w \in \mathcal{A}^d \subset \mathbb{Z}$, *i.e.*,

$$a_t^d = w, \text{ if } w - \frac{W + w}{2W + 1} < \bar{a}_t^d \leq w + \frac{W - w}{2W + 1}.$$

The condition can be written as the range of $a_t^d = w$,

$$\frac{2W + 1}{2W} \left( \bar{a}_t^d - \frac{W}{2W + 1} \right) \leq w < \frac{2W + 1}{2W} \left( \bar{a}_t^d + \frac{W}{2W + 1} \right), \tag{1}$$

and it can be reformulated as

$$w = \min \left( \left\lfloor \frac{2W + 1}{2W} \left( \bar{a}_t^d + \frac{W}{2W + 1} \right) \right\rfloor, W \right),$$

where $\min(\cdot, W)$ hinders $w$ from being outside of action space $[-W, W]$. Here, the floor function is used on the right side of the inequality equation (1). But the ceiling function on the left side of the inequality equation (1) can be an alternative with the max function $\max(\cdot, -W)$.

The post-processor function $a_t^d = g(\bar{a}_t^d, W)$ is finally formulated as follows.

$$g(\bar{a}_t^d, W) = \min \left( \left\lfloor \frac{2W + 1}{2W} \left( \bar{a}_t^d + \frac{W}{2W + 1} \right) \right\rfloor, W \right)$$

## Appendix E    Derivation of (2)

$$\hat{\mathbf{c}}_j = \arg\max_{\mathbf{c}} \ln P(o_{1:T,j}, a_{1:T,j}|\mathbf{c})$$

$$= \arg\max_{\mathbf{c}} \ln \int P(s_{1:T}, o_{1:T,j}, a_{1:T,j}|\mathbf{c}) ds_{1:T} \tag{2}$$

$$= \arg\max_{\mathbf{c}} \ln \int P(s_{1:T}|o_{1:T,j}, a_{t:T,j}) \times \frac{P(s_{1:T}, o_{1:T,j}, a_{1:T,j}|\mathbf{c})}{P(s_{1:T}|o_{1:T,j}, a_{t:T,j})} ds_{1:T} \tag{3}$$

$$= \arg\max_{\mathbf{c}} \int P(s_{1:T}|o_{1:T,j}, a_{t:T,j}) \times \ln \frac{P(s_{1:T}, o_{1:T,j}, a_{1:T,j}|\mathbf{c})}{P(s_{1:T}|o_{1:T,j}, a_{t:T,j})} ds_{1:T} \tag{4}$$

$$= \arg\max_{\mathbf{c}} \int P(s_{1:T}|o_{1:T,j}, a_{t:T,j}) \times \ln P(s_{1:T}, o_{1:T,j}, a_{1:T,j}|\mathbf{c}) ds_{1:T} + H(s_{1:T}|o_{1:T,j}, a_{t:T,j}) \tag{5}$$

$$= \arg\max_{\mathbf{c}} \int P(s_{1:T}|o_{1:T,j}, a_{t:T,j}) \times \ln P(s_{1:T}, o_{1:T,j}, a_{1:T,j}|\mathbf{c}) ds_{1:T} \tag{6}$$

The equality of (2) and (3) is because of multiplying the same value on the numerator and denominator. The inequality of (3) and (4) is based on Jensen's inequality, which means $f(\mathbb{E}[x]) \geq \mathbb{E}[f(x)]$ is satisfied when $f(\cdot)$ is a concave function (in our case, $f(\cdot)$ is $\ln(\cdot)$). Subsequently, we can rewrite $-P(\cdot)\ln P(\cdot)$ as a entropy $H(\cdot)$. The inequality of (5) and (6) is because the entropy $H(\cdot)$ is always a positive value.

$$\hat{\mathbf{c}}_j = \arg\max_{\mathbf{c}} \int P(s_{1:T}|o_{1:T,j}, a_{t:T,j}) \times \ln P(s_{1:T}, o_{1:T,j}, a_{1:T,j}|\mathbf{c}) ds_{1:T}$$

$$= \arg\max_{\mathbf{c}} \int P(s_{1:T}|o_{1:T,j}, a_{1:T,j}) \left[ \ln P(s_1) + \sum_{t=1}^{T} \ln \Omega_j(o_{t,j}|s_t) + \sum_{t=1}^{T} \ln \pi(a_{t,j}|o_{t,j}; \mathbf{c}) \right.$$

$$\left. + \int \sum_{t=1}^{T} \ln \mathcal{T}(s_{t+1}|s_t, a_{t,j}, \mathbf{a}_{t,-j}) d\mathbf{a}_{1:T,-j} \right] ds_{1:T} \tag{7}$$

$$= \arg\max_{\mathbf{c}} \sum_{t=1}^{T} \ln \pi(a_{t,j}|o_{t,j}; \mathbf{c}) \times \int P(s_{1:T}|o_{1:T,j}, a_{1:T,j}) ds_{1:T} \tag{8}$$

$$= \arg\max_{\mathbf{c}} \sum_{t=1}^{T} \ln \pi(a_{t,j}|o_{t,j}; \mathbf{c}) \tag{9}$$

$$= \arg\max_{\mathbf{c}} \sum_{t=1}^{T} [\ln \pi(a_{t,j}^c|o_{t,j}; \mathbf{c}) + \ln \pi(a_{t,j}^d|o_{t,j}; \mathbf{c})] \tag{10}$$

We can decompose (6) as (7) by the Markov property. Next, we can ignore the $\Omega(\cdot)$ and $\mathcal{T}(\cdot)$ of (7) because these terms are not related to $\mathbf{c}$. Likewise, we can ignore the $P(s_{1:T}|o_{1:T,j}, a_{1:T,j})$ of (8). Consequently, (9) can be decomposed as the probabilities with respect to both continuous and discrete action as (10) because we consider the hybrid action space.

## Appendix F  Loss Function for Character Inference

If $\pi(a^c_{t,j}|o_{t,j}; \mathbf{c})$ is the Gaussian distribution and $\pi(a^d_{t,j}|o_{t,j}; \mathbf{c})$ is the Dirac delta distribution, each term of the equation $\mathcal{U}(\mathbf{c})$ is defined as follows:

$$\ln \pi(a^c_{t,j}|o_{t,j}; \mathbf{c}) = \frac{1}{2} \ln 2\pi\sigma^2_\pi + \frac{|a^c_{t,j} - a^{*,c}_{t,j}|}{2\pi\sigma^2_\pi},$$

$$\ln \pi(a^d_{t,j}|o_{t,j}; \mathbf{c}) = \mathbb{1}[a^d_{t,j} \neq a^{*,d}_{t,j}](|a^{*,d}_{t,j} - \bar{a}^d_{t,j}|),$$

where $a^{*,c}_{t,j}$ and $a^{*,d}_{t,j}$ mean the actual action value sampled by observing the target agent, and $\mathbb{1}[\cdot]$ means the indicator function. When the estimated deterministic action $a^d_{t,j}$ is different to the actual action $a^{*,d}_{t,j}$ (i.e., $a^d_{t,j} \neq a^{*,d}_{t,j}$), indicator function becomes 1; Conversely, when $a^d_{t,j} = a^{*,d}_{t,j}$, indicator function becomes 0. If inferred character parameter $\hat{\mathbf{c}}$ is similar to the actual character parameter $\mathbf{c}$, the errors between the action produced by $\hat{\mathbf{c}}$ and the observed actual action would decrease.

# Appendix G  Experiments: Autonomous Driving

To deal with a continuous state space, a hybrid action space, and the agents' characters, we consider the autonomous driving simulator.

In the demonstration task, the agents, the autonomous vehicles, drive the $L$-lane roundabout road. The agents are randomly deployed on the road in every episode. The agents' goal is to drive as close to the desired velocity as possible, and the agents should control the acceleration and lane changes to reach the goal. To address this task, we set the POMDP. Here, the state includes the velocity and position of all vehicles, and the observation includes information about neighboring vehicles. The action includes acceleration and lane change control in continuous and discrete space, respectively. The reward function comprises three terms: considering the desired velocity, safety distance, and meaningless lane change. We provide the specific POMDP model in the following subsection.

## G.1  State

The state $s_t \in \mathcal{S}$ is defined as
$$s_t = [\mathbf{v}_t^T, \mathbf{p}_t^T, \mathbf{k}_t^T]^T.$$
The state $s_t$ means the total information of all vehicles on the road. Here, $\mathbf{v}_t = [v_{t,1}, v_{t,2}, \cdots, v_{t,N}]$ represents the velocity of all vehicles, $\mathbf{p}_t = [p_{t,1}, p_{t,2}, \cdots, p_{t,N}]$ denotes the positions of the vehicles, and $\mathbf{k}_t = [k_{t,1}, k_{t,2}, \cdots, k_{t,N}]$ denotes the lane position of all vehicle at a given time $t$.

## G.2  Observation

The observation $o_t \in \mathcal{O}$ comprises the partial state information that the agent can observe. We assume that an agent $i$ can observe the following and leading vehicles located in the same and next lanes. Thus, we set the observation $o_{t,i}$ as follows:

$$o_{t,i} = [v_{t,i}, \Delta\mathbf{v}_{t,i}, \Delta\mathbf{p}_{t,i}, k_{t,i}]^T,$$

where $v_{t,i}$ denotes the velocity of an agent $i$, $\Delta\mathbf{v}_{t,i}$ is relative velocity between the agent $i$ and observable vehicles, $\Delta\mathbf{p}_{t,i}$ is relative position, and $k_{t,i}$ denotes the lane number at given time $t$. Here, $\Delta\mathbf{v}_{t,i} = [\Delta v_{t,lL}, \Delta v_{t,lS}, \Delta v_{t,lR}, \Delta v_{t,fL}, \Delta v_{t,fS}, \Delta v_{t,fR}]$, and $\Delta\mathbf{p}_{t,i} = [\Delta p_{t,lL}, \Delta p_{t,lS}, \Delta p_{t,lR}, \Delta p_{t,fL}, \Delta p_{t,fS}, \Delta p_{t,fR}]$, where subscripts $l$ and $f$ mean leading and following vehicles, and subscripts $L$, $S$, and $R$ signify located left, same, and right lane, respectively.

## G.3  Action

The action $\mathbf{a}_{t,i} \in \mathcal{A}$ consists of a continuous action $a_{t,i}^c \in \mathcal{A}^c$ and a discrete action $a_{t,i}^d \in \mathcal{A}^d$ at time $t$. In this framework, a continuous action is acceleration control, and a discrete action is a lane change. Acceleration control space $\mathcal{A}^c$ is defined as a space from maximum acceleration to minimum acceleration $[a_{min}, a_{max}]$; Lane change space $\mathcal{A}^d$ is defined as $\{-1, 0, 1\}$. In $\mathcal{A}^d$, $a_{t,i}^d = -1$ means the agent moves a lane outwards (right side), conversely $a_{t,i}^d = 1$ means the agent moves a lane inwards (left side), and $a_{t,i}^d = 0$ means the agent keeps the same lane.

## G.4  Reward

As discussed in section 3.1, the character-based reward function is defined as $R_{t,i} = R_i(s_t, a_{t,i}, s_{t+1}; \mathbf{c}_i)$. In this experiment, the reward function $R_{t,i}$ is defined as:

$$R_{t,i} = c_1\mathcal{R}_1 + c_2\mathcal{R}_2 + c_3\mathcal{R}_3 + r_{fail},$$

where $\mathbf{c} = \{c_1, c_2, c_3\}$ denotes a vector of the character coefficients and $\{\mathcal{R}_1, \mathcal{R}_2, \mathcal{R}_3\}$ denotes a vector of the reward terms, and $r_{fail}$ means a penalty for the unfeasible actions (*i.e.*, trial to move a non-existence lane and a lane where other vehicles are located.).

We use $r_{fail}$ term for punishing unfeasible action, which is designed for safety learning purposes. By introducing this penalty, an agent can learn about unsafe decisions without experiencing an accident. In other words, it allows the agent to use the safety assistant system fewer times, such as the ADAS (Advanced Driver Assistance System).

Subsequently, detailed equations of the reward terms are as follows.

The first reward term is defined as follows:

$$\mathcal{R}_1 = 1 - \left| \frac{v_{t+1,i} - v_i^*}{v_i^*} \right|,$$

where $v_i^*$ denotes the target velocity of the agent $i$. We consider that the agent can drive close to the target velocity. When $v_{t,i} = v_i^*$, the reward term is maximized as the highest value 1; when $v_{t,i} \neq v_i^*$ the reward term is lower than 1.

Next, the second reward term is defined as follows:

$$\mathcal{R}_2(\Delta p_{t+1,fS}) = \min \left[ 0, 1 - \left( \frac{s^*}{\Delta p_{t+1,fS}} \right)^2 \right],$$

where $s^*$ denotes the safety distance between the vehicles, and we design this reward term to induce the agent to drive with the following vehicle in mind when the agent changes the lane. In this reward term, $s^*$ is defined as follows.:

$$s^* = s_0 + \max \left[ 0, v_{t+1,fS} \left( t^* + \frac{\Delta v_{t+1,fS}}{2\sqrt{|A_{min} \times A_{max}|}} \right) \right],$$

where $s_0$ denotes the minimum gap between vehicles, $t^*$ denotes the minimum time headway, the minimum time gap between two sequential vehicles required to arrive at the same location. This safety distance is based on the Intelligent Driving Model (IDM) controller, which is one of the adaptive vehicular control systems [1]. If $s^* \leq \Delta p_{t+1,fS}$ (*i.e.*, the agent keeps the safety distance with a following vehicle when moving the lane), $\mathcal{R}_2$ becomes the 0; on the other hand, $\mathcal{R}_2$ becomes the negative value.

The third term is defined as follows.

$$\mathcal{R}_3 = |a_{t,i}^d| \Delta p_{t,lS} \times \min[0, \Delta p_{t+1,lS} - \Delta p_{t,lS}]$$

This reward term is related to unnecessary lane changes, which is a movement to lanes with less driving space than the current lane. When the agent changes the lane $|a_{t,i}^d| = 1$ and $\Delta p_{t,lS} < \Delta p_{t+1,lS}$ or keeps the lane $|a_{t,i}^d| = 0$, this penalty term can be neglected (*i.e.*, $\mathcal{R}_3 = 0$). Conversely, when the agent changes the lane $|a_{t,i}^d| = 1$ and $\Delta p_{t,lS} \geq \Delta p_{t+1,lS}$, this penalty term becomes the negative value.

# Appendix H    Behavioral Pattern over Character Coefficients

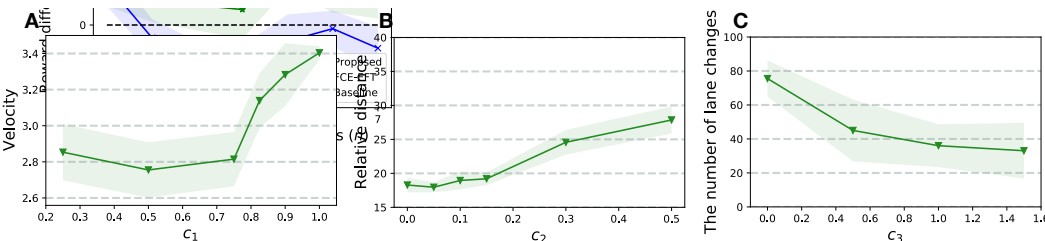

Figure H1: Behavioral pattern of the agent over character coefficient $c_n$. **A**: Tendency of the average velocity of the agent over character $c_1$ ($c_2 = c_3 = 0$). **B**: Tendency of the relative distance to the following vehicle over character $c_2$ ($c_1 = c_3 = 0$). **C**: Tendency of lane-changing frequency over $c_3$ increases($c_1 = c_2 = 0$).

To confirm behavioral differences over the character coefficient, we perform ablation studies on reward function by isolating the independent effect of each character coefficient. It can provide insight into how these characters impact the resulting trajectories. The behavioral differences resulting from character coefficients' changes are illustrated in Figure H1. The markers and shaded areas represent the average value and confidence interval with two standard deviations, respectively.

As described in Appendix G, the reward function is defined as $R_{t,i} = c_1 \mathcal{R}_1 + c_2 \mathcal{R}_2 + c_3 \mathcal{R}_3 + r_{fail}$, where $\mathcal{R}_1$, $\mathcal{R}_2$, and $\mathcal{R}_3$ is related to desired velocity, safe distance and, lane-changing, respectively. Therefore, changes in each character coefficient affect average velocity, relative distance, and the number of lane changes.

Figure H1**A** shows the average velocity of the agent as increasing $c_1$. This result verifies that the autonomous vehicle drives closer to the desired velocity ($v_i^* = 3.5 m/s$). Furthermore, the lower $c_1$ widens the dispersion area of velocity.

Figure H1**B** represents the relative distance between the autonomous vehicle and the surrounding vehicle over $c_2$. The result confirms that the relative distance increases as $c_2$ grows. This character coefficient is straightforwardly related to a safe distance. The agent would pursue safe driving by securing a larger driving space as $c_2$ grows.

Figure H1**C** shows the number of lane changes as $c_3$ increases. In the reward function, $c_3$ puts weights on the unnecessary lane-changing penalty. The unnecessary lane-changing implies movement to lanes with less driving space than the current lane. As $c_3$ decreases, the agent performs lane-chaining action more frequently.

# Appendix I  Performance of Character Inference

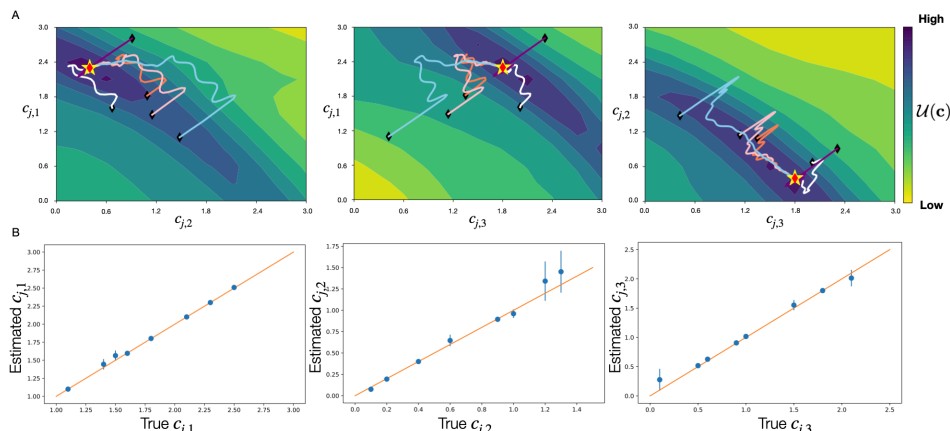

Figure I1: **A**. The converging trajectories of the character parameters. A black diamond indicates the initial points, a red diamond indicates the estimated points, and a yellow star means the true point. **B**. The estimated character parameters of the agent versus true character parameters. The orange line represents the identity line, meaning perfect estimation; the blue circles depict the estimated values, and the blue line presents the confidence interval for three standard deviations.

Figure I1A presents the contour plots of the log-likelihood function for the combination of character parameters $\mathbf{c}_{j,k}$, where $k \in [1, 2, 3]$. It shows that the true value is well inferred no matter where the initial value is located. The yellow star, red and black diamonds in these diagrams represent the true, estimated, and initial points, respectively; the curve line presents the character inference trajectory from an initial point to an estimated point.

Figure I1B shows the estimated character value by the agent $i$ versus the true character value of the target $j$. Each blue point and bar is the average value and the three-standard deviation considering ten experiments. The orange line indicates that the estimated and true values are identical. It represents that the character inference is successful without a large error between the estimated and true value, and in particular, $c_{j,1}$ and $c_{j,3}$ are overall accurate with a small standard deviation. Conversely, the inference about $c_{j,2}$ becomes inaccurate when $c_{j,2} \geq 1.2$. We conclude that the character inference module generally infers the agent's characters well over the observation-action trajectory of the target agent.

# Appendix J   Additional Simulation Results on Autonomous Driving Task

## J.1   Original Plot of Figure 4

Figure J1 shows the original version of Figure 4, *i.e.*, the average reward of entire agents. In a single group scenario (*i.e.*, the entire agents have the same characters), the results of both the proposed and the FCE-EFT solutions are identical. This is because all agents have homogeneous characters, which allows the FCE agent to have the accurate characters of others. The reward of without EFT is lower than two solutions in a single group scenario. This confirms that the proposed EFT mechanism can help the agent to consider multi-agent interactions. Next, in two or more group scenarios, the proposed solution consistently achieves the highest reward, and the FCE-EFT consistently achieves the lowest reward.

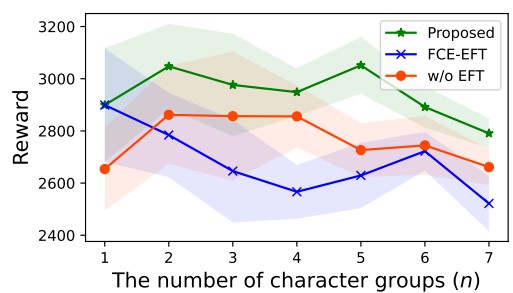

Figure J1: Average reward of entire agents over an increasing number of character groups.

## J.2   Performance Comparison with Confidence Interval

### MARL algorithms:

`MADDPG` [29]: It is a multi-agent version of Deep Deterministic Policy Gradient (DDPG) [27]. In training, it uses a centralized Q-function that uses observations and actions of all agents.

`MAPPO` [63]: It is a multi-agent version of Proximal Policy Optimization (PPO) algorithm [51]. It considers a centralized critic that uses the local observations across all agents.

`Q-MIX` [44]: It uses a mixer and individual $Q$-networks. The mixer network uses the $Q$-values (output of individual $Q$-network) of all agents as inputs and calculates a global $Q_{tot}$ as an output. Since it can only handle the discrete action space, we quantize the continuous actions.

### Model-based RL algorithms:

`Dreamer` [15]: It trains an agent that solves long-horizon tasks purely through latent imagination. This solution first builds a

Table J1: Table 2 with 1 std confidence interval.

| Algorithm | The number of character groups ($n$) | | | | |
| | 1 | 2 | 3 | 4 | 5 |
|---|---|---|---|---|---|
| Proposed | **2899** | **3047** | **2976** | **2948** | **3051** |
| | ±217 | ±162 | ±196 | ±91 | ±109 |
| FCE-EFT | **2899** | 2784 | 2646 | 2566 | 2629 |
| | ±217 | ±161 | ±196 | ±103 | ±125 |
| MADDPG | 2763 | **3006** | 2800 | **2933** | 2856 |
| | ±126 | ±103 | ±106 | ±98 | ±121 |
| MAPPO | 2753 | 2862 | 2597 | 2529 | 2763 |
| | ±206 | ±201 | ±144 | ±131 | ±190 |
| Q-MIX | 2199 | 2310 | 2288 | 2118 | 1861 |
| | ±56 | ±39 | ±118 | ±82 | ±132 |
| Dreamer | **2911** | 2813 | 2733 | 2631 | 2701 |
| | ±312 | ±283 | ±351 | ±521 | ±433 |
| MBPO | 2089 | 1964 | 1523 | 1893 | 1633 |
| | ±804 | ±735 | ±948 | ±792 | ±821 |
| ToMC2 | **3016** | 2812 | 2683 | 2691 | 2511 |
| | ±109 | ±273 | ±309 | ±458 | ±397 |
| LIAM | 1913 | 1792 | 1771 | 1683 | 1733 |
| | ±330 | ±410 | ±367 | ±381 | ±429 |

reward and transition model and then approximates a policy using a value function. This value function is based on leveraging the error between the imagined return and the estimated state value.

`MBPO` [16]: It provides a simple data augmentation process of employing short model-synthesized rollouts branched from the actual trajectory. We train a policy using a blend data, comprising synthesized and actual trajectories.

### Agent modeling algorithms:

`ToMC2` [59]: By incorporating the theory of mind concept, socially intelligent agents are developed that can determine when and to whom they should share their intentions.

`LIAM` [38]: By using autoencoder structures to extract representations from the ego agent's local information, it models the behaviors of other agents in a partially observable environment.

# Appendix K   Multiple Particle Environment

## K.1   Task Description

We select the three MPE tasks: Spread, Adversary, and Tag.

- Spread: In this task, there are three agents. Their objective is to reach three landmarks without collision with each other. A reward function is the sum of negative distances from landmarks to agents and collision penalty term.

- Adversary: This task includes two cooperating agents and a third adversary agent; there are true goal and false goal spots. The adversary can observe relative distances without communication about the goal spots. The cooperative agents aim to reach the goal spot while avoiding an adversary. The reward function is a sum of the negative distance to the goal spot and the distance from the adversary to the true goal. We use an adversary agent controlled by a pre-trained [39].

- Tag: This task is dubbed a predator-prey task. The environment includes two types of agents and obstacles: a single good agent, three adversary agents, and two obstacle blocks. The adversaries are slower than a good agent and receive a reward when tagging a good agent. We employ a pre-trained prey agent from [39].

## K.2   Performance Comparison with Confidence Interval

Table K1: Performance comparison with MARL baseline algorithms.

| Task | MAPPO[†] | MADDPG[†] | Q-MIX[†] | Proposed |
|------|------|------|------|------|
| Spread | $-149.29$ $\pm 0.94$ | $-157.10$ $\pm 2.30$ | $-154.70$ $\pm 4.90$ | $\mathbf{-149.12}$ $\pm 1.38$ |
| Adversary | $9.61$ $\pm 0.07$ | $7.80$ $\pm 1.43$ | $8.11$ $\pm 0.37$ | $\mathbf{10.01}$ $\pm 0.33$ |
| Tag | $13.78$ $\pm 4.40$ | $6.65$ $\pm 3.90$ | $\mathbf{15.00}$ $\pm 2.73$ | $14.57$ $\pm 2.95$ |

# Appendix L    StarCraft Multi-Agent Challenge

## L.1    Task Description

This task includes various scenarios where two armies are controlled by allied agents and the game's AI. Each agent operates under partial observability and can only perceive the environment within its sight range. More precisely, observations include attributes like distance, health, and unit type of nearby allies and enemies. Next, the agents can take actions such as moving in a direction, attacking specific enemies, or healing allies. The objective is to maximize the win rate across episodes. Agents receive rewards based on hit-point damage, enemy kills, and a bonus for winning, while losing results in a negative reward.

We select the three scenarios of the SMAC task: 2s3z, 3s5z_vs_3s6z, and 6h_vs_8z.

- 2s3z: This scenario considers the same number of agents for both alley and enemy. More precisely, each team includes 2 Stalkers and 3 Zealots.
- 3s5z_vs_3s6z: It deploys the 3 Stalkers and 5 Zealots as allies and 3 Stalkers and 6 Zealots as enemies.
- 6h_vs_8z: This combat is performed by 6 Hydralisks against 8 Zealots.

## L.2    Performance Comparison with Confidence Interval

Table L1: Performance comparison with MARL baseline algorithms.

| Task | MAPPO$^\dagger$ | MADDPG | Q-MIX$^\dagger$ | Proposed |
|---|---|---|---|---|
| 2s3z | **100** $\pm1.5$ | 90.3 $\pm5.3$ | 95.3 $\pm2.5$ | 98.8 $\pm2.3$ |
| 3s5z_vs_3s6z | 63.3 $\pm19.2$ | 18.9 $\pm4.8$ | 82.8 $\pm5.3$ | **84.3** $\pm9.1$ |
| 6h_vs_8z | 85.9 $\pm30.9$ | 68.0 $\pm34.7$ | 9.4 $\pm2.0$ | **93.8** $\pm6.7$ |

