# OpenReview forum: "Episodic Future Thinking Mechanism for Multi-agent Reinforcement Learning"
_NeurIPS.cc/2024/Conference — NeurIPS 2024 poster_

### Official Review · Reviewer_ukHS · 2024-06-19

**Soundness:** 2
**Presentation:** 3
**Contribution:** 3
**Rating:** 6
**Confidence:** 3

**Summary:**

This paper introduces an Episodic Future Thinking (EFT) mechanism for reinforcement learning (RL) agents to enhance decision-making in multi-agent scenarios. The EFT mechanism allows an agent to predict the future actions of other agents by inferring their characters from observation-action trajectories. This capability is evaluated in multi-agent autonomous driving scenarios and multiple particle environments, demonstrating that EFT leads to higher rewards compared to existing multi-agent RL solutions.

**Strengths:**

The integration of episodic future thinking in RL is a significant contribution, providing a new perspective on how agents can predict and simulate future scenarios to improve decision-making. Besides, The paper is well-structured and clearly explains the methodology, experiments, and results, also provides comprehensive evaluations in diverse experiments, including ablation study, showcasing the robustness of the proposed method.

**Weaknesses:**

1.	The paper does not sufficiently address the computational overhead of implementing the EFT mechanism, especially with varying data sizes.
2.	I suggest that authors also implement SOTA in the experiment of investigating the effects of trajectory noise, so that compare the sensitivity of the proposed methods.
3.	The approach assumes that character traits can be inferred accurately, which might not hold in highly dynamic environments with rapidly changing behaviors.

**Questions:**

1. How does the performance of the proposed Episodic Future Thinking mechanism scale with an increasing number of agents?

**Limitations:**

The approach assumes that character traits can be inferred accurately, which might not hold in highly dynamic environments with rapidly changing behaviors. The limitation of having only one EFT-enabled agent in experiments raises questions about the method's effectiveness in scenarios where multiple agents are equipped with EFT capabilities.

---

> ### Author Rebuttal · Authors · 2024-08-07
>
> We thank the reviewer for the positive reviews and insightful feedback about this work. Below, we describe how we have revised the paper to address the reviewer's concerns and questions.
> - **Computational complexity**
>
> We agree that considering computational complexity is crucial for practical solution development. To address the reviewer's concern, we have investigated it using a big $\mathcal O$ analysis of the proposed solution with our setup. Below are the notations used in this analysis:
> 1. $d$: denote the dimension of the input
> 2. $|E|$: the number of agents
> 3. $|E_{obs}|$: the number of observable agents
>
> Before looking at the specific analysis, \textbf{the complexities of EFTM is $\mathcal O(|E_{obs}|\times d^2)$ for the execution}, and vanilla policy requires $\mathcal O(d^2)$. This implies that **the maximum time complexity of EFTM is limited, regardless of the environment's size**, since the maximum number of observable agents is fixed.
>
> We provide how to calculate the complexity of basic policy operations as shown in the below table. This demonstrates that the complexity of these operations is $\mathcal{O}(d^2)$. For EFT prediction, our solution requires $|E_{obs}|$-times the computational cost for others' action prediction. Therefore, the complexity of the proposed solution is $\mathcal O(|E_{obs}|\times d^2)$ for the execution.
> |Computation|Equation|Matrix Size|Complexity|
> |-|-|-|-|
> |The $1^{\mathrm{st}}$ policy layer|$\mathrm{out}_1 = \sigma_1(W_1\cdot x_t+b_1)$|$W_1 \in \mathbb{R}^{2d\times d}, x_t \in \mathbb{R}^{d}$|$2d^2$|
> |The $2^{\mathrm{st}}$ policy layer|$\mathrm{out}_2 = \sigma_2(W_2\cdot \mathrm{out}_1+b_2)$|$W_2 \in \mathbb{R}^{4d\times 2d}, \mathrm{out}_1 \in \mathbb{R}^{2d}$|$8d^2$|
> |The output layer|$a_t = \tanh(W_3\cdot\mathrm{out}_2 + b_3)$|$W_3 \in \mathbb{R}^{2 \times 4d}, \mathrm{out}_2 \in \mathbb{R}^{4d}$|$8d$|
> |**Total of policy**|-|-|$\mathcal{O(d^2)}$|
>
> Our solution focuses on predicting the actions of agents within the observation area, not the entire state. It means that regardless of the environment's size, the number of observable agents is limited. These computations can be performed in parallel, making our analysis more relevant and effective than simply comparing execution times for single operations.
>
> Next, our solution has advantages in terms of training time. To elaborate, the wall-clock times of EFTM and MARL (averaging MAPPO, MADDPG, and QMIX) are as follows: **for autonomous driving tasks, approximately $3.5$ and $17$ hours, and for MPE tasks, approximately $1.7$ and $2.5$ hours**. This significant gap comes from our objective, training a multi-character policy that can work in any multi-agent interactions. It means that EFTM considers training a single agent, not multi-agents.
>
> ---
> - **Comparison of the trajectory noise sensitivity with baseline**
>
> Thank you for your comments, which helped us to improve the experimental section. To address the reviewer's concern, we have run additional experiments. Given time constraints, we only performed additional investigation on MADDPG, which showed the second-best performance in our selective demonstration tasks. Below are additional results ($4$ seeds).
> |std of trajectory noise|0.0|0.01|0.05|0.1|0.2|0.3|
> |-|-|-|-|-|-|-|
> |MADDPG (test noise)|2763 $\pm$ 126|2530 $\pm$ 439|1891 $\pm$ 892|1522 $\pm$ 1039|837 $\pm$ 711|335 $\pm$ 693|
> |MADDPG (training noise)|2763 $\pm$ 126|2891 $\pm$ 360|2610 $\pm$ 402|2133 $\pm$ 519|1258 $\pm$ 1011|1341 $\pm$ 955|
> |EFTM|2899 $\pm$ 217|2833 $\pm$ 316|2841 $\pm$ 283|2795 $\pm$ 613|2437 $\pm$ 812|1535 $\pm$ 1023|
>
> The empirical result confirms that **the proposed solution is more robust than the MADDPG**. We conjecture that the proposed solution might alleviate noise effects through character classification since noise information is not used for direct action computation.
>
> Details of two types of baselines are as follows.
> 1. Test noise without training noise: This case adds the noise into the MARL agents' observation directly. Direct noise for policy calculations quickly leads to the collapse of the entire multi-agent system.
> 2. Training noise without test noise: This case adds the trajectory noise in a training phase, \textit{i.e.}, building policy and team value function. A small amount of noise during the learning process serves to increase the robustness of the overall system, but as the noise increases, the instability of the learning process increases.
>
> Please be aware that the considered standard deviation is not trivial given that our observation range is $[-1, 1]$. Specifically, we provide the signal-to-noise ratio with a quality level across each standard deviation. We label the quality level from [15].
> |std of trajectory noise|0.01|0.05|0.1|0.2|0.3|
> |-|-|-|-|-|-|
> |signal-to-noise ratio|34.7dB|21.3dB|14.7dB|9.2dB|4.7dB|
> |quality level|Excellent|Good|Fair|Poor|Poor|

---

> > ### Comment · Reviewer_ukHS · 2024-08-09
> > **Thank you for the responses**
> >
> > Thank you to the authors for their responses. Most of my questions have been addressed. After considering your responses and the feedback from other reviewers, I will maintain my evaluation.

---

> > > ### Author Response · Authors · 2024-08-10
> > >
> > > Thank you once again for your active engagement and for taking time and effort into this discussion!

---

> ### Author Response · Authors · 2024-08-07
>
> - **Limitation in highly dynamic environments with rapidly changing behaviors**
>
> We appreciate the reviewer's insightful comment regarding the challenges of modeling and inference with policy changes over time. As the EFT agent should continuously adapt to evolving strategies and behaviors, the complexity of modeling and inferring these changes increases significantly. This issue is further compounded as the number of agents grows, potentially exacerbating the intractability of the problem. The dynamic nature of policy introduces additional layers of complexity, making it increasingly difficult to predict and manage the interactions among agents effectively.
>
> This is an ultimate goal for the research community, and we consider it as future work. Although we did not address rapidly changing behavior in this study, our work demonstrates promising results, such as successful interactions with changes in surrounding characters across different episodes. In accordance with NeurIPS policy, we would like to clarify that we have already discussed this limitation in our manuscript.
>
> ---
>
> Once again, we deeply appreciate the insightful comments and suggestions. We hope our clarification and additional empirical studies could address the concerns raised by the reviewer. Should there be any leftover questions, please let us know and we will make every effort to address them during the subsequent discussion period.

---

### Official Review · Reviewer_6v5k · 2024-06-25

**Soundness:** 2
**Presentation:** 3
**Contribution:** 2
**Rating:** 5
**Confidence:** 2

**Summary:**

Introduce an episodic future thinking(EFT) mechanism, which, along with the mechanism of counterfactual, is a cognitive activity of human beings.
The proposed algorithm predicts future observation transitions and uses them to determine the next steps of action. Although the maximum likelihood method is also used to infer a character c, I do not believe that this paper has made a significant contribution.

**Strengths:**

Introduce an episodic future thinking(EFT) mechanism, which, along with the mechanism of counterfactual, is a cognitive activity of human beings.
The proposed algorithm predicts future observation transitions and uses them to determine the next steps of action. Although the maximum likelihood method is also used to infer a character c, I do not believe that this paper has made a significant contribution.

**Weaknesses:**

The proposed algorithm predicts future observation transitions and uses them to determine the next steps of action. Although the maximum likelihood method is also used to infer a character c, I do not believe that this paper has made a significant contribution.

**Questions:**

No

---

> ### Author Rebuttal · Authors · 2024-08-07
>
> We appreciate the reviewer's time and effort. Here are our answers to the reviewer's comments.
>
> - **Our contribution and motivation**
>
> We would like to clarify our contribution is not trivial. To emphasize our contribution, we summarize the novelty as follows. **We introduce a novel social decision-making approach by coupling character inference and upcoming future prediction**. Each independent functionality has been studied partially in different research streams but never jointly considered for social decision-making scenarios. **Moreover, its effectiveness in a scenario where heterogeneous agents coexist has never been experimentally proved**. Combining such functionality in the social decision-making framework is not straightforward or incremental, which is why we put considerable effort into clearly explaining the proposed solution.
>
> Additionally, all other reviewers acknowledged that our method is novel with detailed points:
> - Reviewer Cur7: A policy that can handle diverse agent characters is a significant contribution. This addresses a real challenge in multi-agent systems where agents may have different goals or behavioral traits.
> - Reviewer ukHS: The integration of episodic future thinking in RL is a significant contribution, providing a new perspective on how agents can predict and simulate future scenarios to improve decision-making.
> - Reviewer RgnJ: The multi-character policy handles both continuous and discrete action spaces, expanding the applicability of RL methods to more complex scenarios.
> - Reviewer pTth: The cognitive motivation makes a lot of sense, and broadly, modeling diverse other agent motives seems like a promising direction that has not received much attention.
>
> This work addresses a significant topic in the advancement of the MARL domain. As the reviewer cur7 mentioned, handling diverse characters is a substantial challenge in multi-agent systems where agents may have different goals or behavioral traits.
>
> While we sincerely want to provide more detailed responses, there is limited information about the discussion points, so we could not elaborate further. Please feel free to ask any additional questions the reviewer may have, and we will be happy to answer them.

---

> > ### Author Response · Authors · 2024-08-12
> >
> > Thank you for raising the scores. We confirmed that the reviewer changed scores from 4 to 5. If the reviewer could provide an opinion on what additional work is needed for us to move beyond the borderline score, we would greatly appreciate it!

---

### Official Review · Reviewer_Cur7 · 2024-07-12

**Soundness:** 3
**Presentation:** 3
**Contribution:** 3
**Rating:** 6
**Confidence:** 4

**Summary:**

This paper introduces an Episodic Future Thinking (EFT) mechanism for reinforcement learning agents in multi-agent systems with heterogeneous characters. The authors propose a multi-character policy and a character inference module to enable agents to predict other agents' actions and simulate future scenarios. The EFT mechanism allows agents to make adaptive decisions by considering the predicted future state. The approach is evaluated in autonomous driving scenarios and multiple particle environments, demonstrating improved performance compared to existing multi-agent and model-based reinforcement learning algorithms.

**Strengths:**

-  A policy that can handle diverse agent characters is a significant contribution. This addresses a real challenge in multi-agent systems where agents may have different goals or behavioral traits.
- The authors test their approach across various levels of character diversity and compare it with multiple reasonable baselines. This thorough evaluation strengthens the validity of their claims.
- The method's effectiveness is demonstrated in both autonomous driving and multiple particle environments, suggesting potential applicability across different domains.
- The paper is generally well-structured and clearly written.

**Weaknesses:**

-  The experiments only consider one EFT agent among non-EFT agents.
- The results in Figure 4 are not statistically significant. There is also no standard deviation for the baseline.
- No standard deviations provided in Table 2 and 3.
- The difference between training and execution wasn't clear until it was mentioned in the conclusion.
- The paper lacks a detailed analysis of the computational costs associated with the EFT mechanism, particularly as the number of agents or environmental complexity increases.
- While the paper mentions POMDP, it doesn't deeply explore how partial observability affects the performance of the EFT mechanism.
-  The improvement over baseline methods, while present, is not consistently substantial across all scenarios, particularly in the multiple particle environments.

**Questions:**

-  “In contrast, our solution trains the policy with only local observations and actions, which can be a more practical solution.” But you still need to train the character identification model and multi-character policy, which requires access to the other observations too?
- “In addition, the standard deviation of model-based RL algorithms is much larger than the proposed solution, which shows the difficulty of learning a dynamic model without understanding others in multi-agent systems.“. What standard deviations are the authors referring to?
- How does the EFT mechanism perform when all agents in the system are equipped with this capability? Does this lead to emergent behaviors or potential instabilities?
- What is the scalability of the proposed method? How does its performance and computational cost change as the number of agents increases?
- How robust is the character inference module to noisy or adversarial behaviors from other agents?

**Limitations:**

- The current study only considers scenarios with a single EFT agent, which doesn't fully capture the dynamics of multiple predictive agents interacting. In scenarios where multiple agents use EFT, there's a potential for feedback loops or cascading effects that could lead to suboptimal or unstable system behavior. This isn't explored in the current work.
- The paper doesn't address the potential increased computational requirements of the EFT mechanism compared to simpler approaches, which could be a limitation in resource-constrained environments. A table or figure comparing wall-clock time would be insightful.
- While the method is tested in two different environments, its performance in more complex, dynamic, or partially observable environments remains unexplored.

This paper presents an interesting approach to multi-agent reinforcement learning by incorporating episodic future thinking. While the idea is novel and shows some promise, the lack of statistically significant increase in performance and lack of comparison with regards to wall-clock time in the current study, raise concerns about its broader applicability and impact.

---

> ### Author Rebuttal · Authors · 2024-08-07
>
> We are thankful for the reviewer’s detailed feedback and constructive suggestions for improving our work. In response, we outline the revisions made to address the reviewer’s concerns and questions. We have marked the weakness, question, and limitation numbers associated with each discussion section.
> - **Computational complexity (W5, Q4, L2)**
>
> We agree that considering computational complexity is crucial for practical solution development. To address the reviewer's concern, we have investigated it using a big $\mathcal O$ analysis of the proposed solution with our setup. Below are the notations used in this analysis:
> 1. $d$: denote the dimension of the input
> 2. $|E|$: the number of agents
> 3. $|E_{obs}|$: the number of observable agents
>
> Before looking at the specific analysis, \textbf{the complexities of EFTM is $\mathcal O(|E_{obs}|\times d^2)$ for the execution}, and vanilla policy requires $\mathcal O(d^2)$. This implies that **the maximum time complexity of EFTM is limited, regardless of the environment's size**, since the maximum number of observable agents is fixed.
>
> We provide how to calculate the complexity of basic policy operations as shown in the below table. This demonstrates that the complexity of these operations is $\mathcal{O}(d^2)$. For EFT prediction, our solution requires $|E_{obs}|$-times the computational cost for others' action prediction. Therefore, the complexity of the proposed solution is $\mathcal O(|E_{obs}|\times d^2)$ for the execution.
> |Computation|Equation|Matrix Size|Complexity|
> |-|-|-|-|
> |The $1^{\mathrm{st}}$ policy layer|$\mathrm{out}_1 = \sigma_1(W_1\cdot x_t+b_1)$|$W_1 \in \mathbb{R}^{2d\times d}, x_t \in \mathbb{R}^{d}$|$2d^2$|
> |The $2^{\mathrm{st}}$ policy layer|$\mathrm{out}_2 = \sigma_2(W_2\cdot \mathrm{out}_1+b_2)$|$W_2 \in \mathbb{R}^{4d\times 2d}, \mathrm{out}_1 \in \mathbb{R}^{2d}$|$8d^2$|
> |The output layer|$a_t = \tanh(W_3\cdot\mathrm{out}_2 + b_3)$|$W_3 \in \mathbb{R}^{2 \times 4d}, \mathrm{out}_2 \in \mathbb{R}^{4d}$|$8d$|
> |**Total of policy**|-|-|$\mathcal{O(d^2)}$|
>
> Our solution focuses on predicting the actions of agents within the observation area, not the entire state. It means that regardless of the environment's size, the number of observable agents is limited. These computations can be performed in parallel, making our analysis more relevant and effective than simply comparing execution times for single operations.
>
> Next, our solution has advantages in terms of training time. To elaborate, the wall-clock times of EFTM and MARL (averaging MAPPO, MADDPG, and QMIX) are as follows: **for autonomous driving tasks, approximately $3.5$ and $17$ hours, and for MPE tasks, approximately $1.7$ and $2.5$ hours**. This significant gap comes from our objective, training a multi-character policy that can work in any multi-agent interactions. It means that EFTM considers training a single agent, not multi-agents.
>
> ---
> - **Accessibility of others' trajectories (W6, Q1, Q5)**
>
> Thank you for the thorough review and insightful comments. In this response, we would like to address the accessibility assumption in MARL and alternatives in terms of POMDP setup.
>
> (1) *Accessibility assumption*
>
> We would like to explain the details of multi-character policy training and our consideration in the character inference process.
>
> **We build a multi-character policy without the accessibility of other's observations**. For training the multi-character policy, we follow this process:
> 1. Sample the character for an EFT agent. This step allows the agents to experience various characters. Each character component can be sampled from their pre-defined ranges.
> 2. Episode runs and train policy based on the RL process.
> 3. Iterative 1-2 steps at every episode.
>
> Regarding the character inference process, **while it requires the other's observation, this does not imply direct access to the other's observation**. The agent can collect by observing other's trajectories. To address such practical concerns, we have studied the performance robustness corresponding to the trajectory noise level in the below discussion.
>
> (2) *Robustness against trajectory noise of POMDP setup*
>
> We would like to clarify that the **one core aspect of partially observable MDP is the noisy level [14]**. We have provided a study of how robust our solution is as the noise level of the observed trajectory increases in our original manuscript (Section 5.2). In addition, we add the performance of EFTM matching with each standard deviation of additive Gaussian noise, as follows.
> |Std of additive Gaussian noise|0.01|0.05|0.1|0.2|0.3|
> |-|-|-|-|-|-|
> |Inference Accuracy|99.6 $\pm$ 0.01|98.3 $\pm$ 0.07|91.8 $\pm$ 0.23|81.1 $\pm$ 0.52|69.5 $\pm$ 0.66|
> |Cumulative reward of EFTM|2833 $\pm$ 316|2841 $\pm$ 283|2795 $\pm$ 613|2237 $\pm$ 812|1435 $\pm$ 1023|
>
> We believe that this result provides valuable insights into the expected performance of the proposed solution, particularly in scenarios where observation prediction technology is deployed. Although we only consider noise level without an adversarial agent scenario, we politely assert that such a scenario is beyond our scope.
>
> Please be aware that the considered standard deviation is not trivial given that our observation range is $[-1, 1]$. Specifically, we provide the signal-to-noise ratio with a quality level across each standard deviation. We label the quality of each level based on [15].
> |std of trajectory noise|0.01|0.05|0.1|0.2|0.3|
> |-|-|-|-|-|-|
> |signal-to-noise ratio|34.7dB|21.3dB|14.7dB|9.2dB|4.7dB|
> |quality level|Excellent|Good|Fair|Poor|Poor|

---

> ### Author Response · Authors · 2024-08-07
>
> - **Experimental results as the number of EFT agents increases (W1, L1)**
>
> Although we have discussed this potential weakness in our limitation section, we additionally explored how our EFTM behaves as the number of agents increases. The additional empirical results are as follows.
> Here's the table with the bold formatting removed:
> |Ratio of EFT agent|Baseline (single EFT)|10%|20%|30%|40%|50%|60%|
> |-|-|-|-|-|-|-|-|
> |Performance|2899 $\pm$ 217|2910 $\pm$ 193|2818 $\pm$ 316|2376 $\pm$ 991|2041 $\pm$ 752|1650 $\pm$ 548|1728 $\pm$ 683|
>
> Empirical result indicates that EFTM performance remains robust when the interaction between EFT agents is infrequent, such as around $20\%$, and that performance gradually declines thereafter. As per our expectations, potential instability happens when a larger proportion of agents in the system are equipped with EFT simultaneously.
>
> This is similar to the ongoing debates in the theory of mind (ToM) research, where the complexity and depth of understanding others' mental states—from zero- to higher-order ToM—are crucial points of discussion. Determining the optimal level of complexity for specific scenarios is an interesting direction for future research and could offer valuable insights into EFTM.
>
> ---
> - **Standard deviation for main results and additional experiments on SMAC (W3, W7, Q2, L3)**
>
> We apologize for the inconvenience. We wanted to report it in the main body, but due to the page limit, we included it in the appendix of the original manuscript. Our appendix includes Tables with the standard deviation as follows.
> |Character diversity|n=1|n=2|n=3|n=4|n=5|
> |-|-|-|-|-|-|
> |Proposed|**2899** $\pm$ 217|**3047** $\pm$ 162|**2976** $\pm$ 196|**2948** $\pm$ 91|**3051** $\pm$ 109|
> |FCE-EFT|**2899** $\pm$ 217|2784 $\pm$ 161|2646 $\pm$ 196|2566 $\pm$ 103|2629 $\pm$ 125|
> |MADDPG|2763 $\pm$ 126|**3006** $\pm$ 103|2800 $\pm$ 106|**2933** $\pm$ 98|2856 $\pm$ 121|
> |MAPPO|2753 $\pm$ 206|2862 $\pm$ 201|2597 $\pm$ 144|2529 $\pm$ 131|2763 $\pm$ 190|
> |QMIX|2199 $\pm$ 56|2310 $\pm$ 39|2288 $\pm$ 118|2118 $\pm$ 82|1861 $\pm$ 132|
> |Dreamer|**2911** $\pm$ 312|2813 $\pm$ 283|2733 $\pm$ 351|2631 $\pm$ 521|2701 $\pm$ 433|
> |MBPO|2089 $\pm$ 804|1964 $\pm$ 753|1523 $\pm$ 948|1893 $\pm$ 792|1633 $\pm$ 821|
>
> |Algorithm|MAPPO|MADDPG|QMIX|Proposed|
> |-|-|-|-|-|
> |Spread|-149.29 $\pm$ 0.94|-157.10 $\pm$ 2.30|-154.70 $\pm$ 4.90|**-149.12** $\pm$ 1.38|
> |Adversary|9.61 $\pm$ 0.07|7.80 $\pm$ 1.43|8.11 $\pm$ 0.37|**10.01** $\pm$ 0.33|
> |Tag|13.78 $\pm$ 4.40|6.65 $\pm$ 3.90|**15.00** $\pm$ 2.73|14.57 $\pm$ 2.95|
>
> These tables show that **the standard deviation of EFTM is similar to that of other methods**. The model-based solution has the highest variance due to the uncertainty of other agents. Overall, EFTM achieves the best performance with a mid-level variance compared to all other baselines.
>
> Additionally, we have run additional experiments on SMAC [1], which is widely used for evaluating the MARL algorithm, to address the reviewer's concern. We report the performance ($4$ seeds) with MARL baselines, as follows.
> |SMAC Task|EFTM|MAPPO|MADDPG|QMIX|
> |-|-|-|-|-|
> | 2s3z|98.8 $\pm$ 2.3| **100** $\pm$ 1.5|90.3 $\pm$ 5.3|95.3 $\pm$ 2.5|
> | 3s5z vs 3s6z |**84.3** $\pm$ 9.1|63.3 $\pm$ 19.2|18.9 $\pm$ 4.8|82.8 $\pm$ 5.3|
> | 6h vs 8z|**93.8** $\pm$ 6.7|85.9 $\pm$ 30.9|68.0 $\pm$ 34.7|9.4 $\pm$ 2.0|
> | Total|**276.9**|249.2|177.2|187.5|
>
> This result also demonstrates that EFTM still has surpassing or matching performance with previous solutions. It means that EFTM is capable of generalizing to solve widely-used MARL tasks, achieving the best total scores. Notably, we set a simple setup for the SMAC and MPE environments that is, we follow a vanilla setup with a single character diversity $n=1$. **While fully leveraging the advantages of EFTM in these environments can be challenging, EFTM is nonetheless capable of delivering competitive performance in such settings.**
>
> ---
> Once again, we deeply appreciate the insightful comments and suggestions. We hope our clarification and additional empirical studies could address the concerns raised by the reviewer. Should there be any leftover questions, please let us know and we will make every effort to address them during the subsequent discussion period.

---

> > ### Comment · Reviewer_Cur7 · 2024-08-07
> >
> > I thank the authors for running additional experiments and addressing my weaknesses. The additional results improved my outlook on the paper!
> >
> > I have one immediate follow-up question for clarification. In your experiments, for example SMAC, does the character inference process get the observations directly or does the EFT agents collect them, as proposed in your new ablation?
> >
> > Furthermore, I appreciate the new table results with standard deviations. I believe all means should be boldened where the standard deviations overlap for the final version of the paper.

---

> ### Author Response · Authors · 2024-08-07
>
> Thank you for your active response! As discussed earlier, we set character diversity as n=1 on SMAC and MPE. It means that the EFT agent does not need to infer the character because they have the same; in addition, the EFT agent only predicts teammates’ future actions, not including opponents. Additional experiments aim to study whether EFTM-based action selection works in other environments.
>
> Next, we promise to follow the reviewer’s suggestion about performance highlighting style.

---

> > ### Comment · Reviewer_Cur7 · 2024-08-08
> >
> > Thanks for the quick response.
> >
> > Given the rebuttal and the additional results, I will update my score to a 6, expecting a moderate-to-high-impact.
> > I find the empirical evaluation solid and interesting to the community. The combination of components is unique to the best of my knowledge. I believe it is interesting to the field that this combination of components is valuable and the analyses highlight further limitations and lays the ground for future work.
> >
> > I do not think the performance of the algorithm justifies a 7, expecting high-impact. For example, in SMAC, the proposed method performs as well as MAPPO or MADDPG, even at high character diversity (n=5), when accounting for the standard deviations, which itself is a fair baseline but also not necessarily state-of-the-art. Similar conclusions hold for MPE. However, given the improved wall-clock time and different training regime, this is still a significant contribution. Realistically, for high impact, the performance improvements would probably need to be better to motivate a large subgroup of the field to improve on this method.

---

> > > ### Author Response · Authors · 2024-08-08
> > >
> > > We sincerely appreciate the insights you’ve shared for this work and are truly grateful for raising the score. Your detailed explanation regarding the score update is extremely helpful.
> > >
> > > As for SMAC performance, we could not fully explore the hyperparameters due to the limited rebuttal time. Moving forward, we will make more effort to have a higher impact!
> > >
> > > Thank you once again for your active engagement in this discussion. We truly appreciate the time and effort you’ve dedicated!

---

### Official Review · Reviewer_pTth · 2024-07-13

**Soundness:** 3
**Presentation:** 3
**Contribution:** 2
**Rating:** 6
**Confidence:** 4

**Summary:**

This paper presents Episodic Future Thinking (EFT), an approach for RL in multi-agent environments. EFT involves learning a multi-character policy (where character is a parameter that modifies the reward), and then using this to infer characters of other agents and planning accordingly, using these characters and learned policy to predict others’ trajectories more accurately. The paper demonstrates superior performance on a driving environment and multi-agent particle environments.

**Strengths:**

The paper is clearly written throughout. It presents, to my knowledge, an original approach for multi-agent RL with characters. Results are well-described and make sense. The studies of 5.2 and 5.3 are welcome additions that help make sense of how the method works. The cognitive motivation makes a lot of sense, and broadly, modeling diverse other agent motives seems like a promising direction that has not received much attention.

*Edit*: raised score to 6 following rebuttal.

**Weaknesses:**

My main concern is with the significance of the performance comparisons. For the driving task, my understanding is that the other agents have a range of characters. The proposed method has the opportunity to learn a multi-character policy. First, I have a concern as to how one might put the baselines on an equal footing in terms of experience — see Questions for that. Second, even if the baselines were put on an equal footing in terms of experience, how surprising is the result for the driving experiment? The driver environment has been designed so that the proposed method has precisely the right inductive bias — inferring a latent character vector.

The MPE testbed is less clearly set up so that the proposed model has the right inductive bias for it — though perhaps it helps to be able to have separate models of the different agent groups — and again (see Questions), it’s really unclear to me how you would put baselines on the same footing in terms of giving them experience modeling both groups.

It would be very helpful to include confidence intervals for these experiments. Performance on MPE testbed is very close, numerically, to baselines. Are those differences actually statistically significant? I think those environments tend to have pretty high variance.

The model-based baselines, especially Dreamer, shouldn’t be expected to work well in multi-agent environments like these without significant modifications, I think. Dreamer is not going to handle stochasticity of multi-agent environments well well given how the world model is set up by default. Did you modify it? And why use Dreamer v1 instead of the most recent version?

**Questions:**

Given that the proposed method gets to train a multi-character policy, which presumably involves training on a bunch of experience with multiple characters, how are the baselines put on an equal footing in terms of experience in the environment, with these different character objectives?

How is c varied during multi-character policy training? Is it randomly set each episode?

Minor, and I may have missed this, but what model is used to do forward prediction? It might be helpful to briefly mention that in the main text, if it’s not there.

**Limitations:**

Seems adequate, if the above are addressed.

---

> ### Author Rebuttal · Authors · 2024-08-07
>
> We are grateful for the reviewer's thorough review and valuable suggestions about this work. Below, we outline how we have revised the paper to address the reviewer's concerns and questions.
> - **Fairness for experience in character diversity**
>
> We agree with the reviewer that maintaining a fair experimental setup is crucial for performance comparison. As the reviewer correctly pointed out, we set up MARL agents with a single character during their training. This is because we thought character change could impede MARL training.
>
> To address the reviewer's comment and confirm our belief, **we provide additional evaluations of baselines with a diverse character experience** - an equal footing in terms of character experience. We report the results (4 seeds) below.
> |Algorithm|n=1|n=2|n=3|n=4|n=5|
> |-|-|-|-|-|-|
> |EFTM|2899 $\pm$ 217|3047 $\pm$ 162|2976 $\pm$ 196|2948 $\pm$ 91|3051 $\pm$ 109|
> |**MADDPG (new)**|2368 $\pm$ 85|2419 $\pm$ 262|2353 $\pm$ 56|2310 $\pm$ 142|2338 $\pm$ 210|
> |MADDPG (original)|2763 $\pm$ 126|3006 $\pm$ 103|2800 $\pm$ 106|2933 $\pm$ 98|2856 $\pm$ 121|
> |**MAPPO (new)**|2192 $\pm$ 113|2366 $\pm$ 91|2233 $\pm$ 250|2180 $\pm$ 313|2241 $\pm$ 386|
> |MAPPO (original)|2753 $\pm$ 206|2862 $\pm$ 201|2597 $\pm$ 144|2529 $\pm$ 131|2763 $\pm$ 190|
>
> This result confirms that EFTM outperforms other solutions in a demonstration task. As same with our expectation, **other baselines show a performance degradation compared to the existing setup**. Nevertheless, the performance is maintained rather than decreased as the character diversity level $n$ increases. We conjecture its performance drop is caused by not considering the optimization method for multi-characters. Although we provide the experience variability to agents, they cannot use it efficiently.
>
> Here are explanations of additional procedures for ensuring a fair experience.
> 1. Sample the level of character diversity in society. This step allows the agents to experience various levels of society. We establish the set of character diversity levels as $[1, 2, 3, 4, 5]$, considered in the evaluation phase.
> 2. Sample the character for each agent. This step allows the agents to experience various characters. Each character component can be sampled from their pre-defined ranges.
> 3. Episode runs.
> 4. Iterative 1-3 steps.
>
> ---
> - **Simulation detail for MPE task**
>
> We would like to explain the detailed setup for additional tasks. For the MPE task, we consider a simple setup, akin to prior works, with character as a single character, that is, the diversity level $n=1$. For competitive tasks, we deployed the EFT agent on a good agent group and the pre-trained networks on an adversarial one. The EFT agent only predicts teammates' future actions, not an adversarial group.
>
> Please note that the main body and appendix of the original manuscript include experimental setup information as described below.
>
> **Main body**: We set the character for each group as a single character, that is, the diversity level $n=1$.
>
> **Appendix**:
> 1. Spread: In this task, there are three agents. Their objective is to reach three landmarks without collision with each other. A reward function is the sum of negative distances from landmarks to agents and collision penalty term.
> 2. Adversary: This task includes two cooperating agents and a third adversary agent; there are true goal and false goal spots. The adversary can observe relative distances without communication about the goal spots. The cooperative agents aim to reach the goal spot while avoiding an adversary. The reward function is a sum of the negative distance to the goal spot and the distance from the adversary to the true goal. We use an adversary agent controlled by a pre-trained [7].
> 3. Tag: This task is dubbed a predator-prey task. The environment includes two types of agents and obstacles: a single good agent, three adversary agents, and two obstacle blocks. The adversaries are slower than a good agent and receive a reward when tagging a good agent. We employ a pre-trained prey agent from [7].
>
> ---
> - **Version of Dreamer**
>
> Thank you for providing key discussion points about the proper baseline selection. We initially considered Dreamer v1 [8], v2 [9], and v3 [10] as candidates for baselines, and we selected Dreamer v1 for the following reasons. Among these, **Dreamer v3 has not been peer-reviewed yet, so we decided not to use it**. Next, we delved into Dreamer v1 and v2 papers, and then we found that they focused on different demonstration tasks. **Dreamer v2 is more focused on discrete control tasks**, for example, the Atari games [11]. Conversely, Dreamer v1 concentrated on continuous control tasks, such as DeepMind control suite [12], DeepMind lab [13], and some continuous Atari games [11].
>
> Subsequently, when considering model-based baselines, we only train a single agent with a world model in the training phase. We then deploy trained agents in driving environments, where there are pre-trained drivers with diversified characters. Note that deployed agents in the training and test phases are the same as EFTM. In summary, we did not over-modify the existing Dreamer model.
>
> ---
> - **How does the character $c$ determine?**
>
> As the reviewer understands, **we randomly sample the character $\mathbf {c}$ from a character distribution in every episode**. More precisely, we use a uniform sampling method to set the character of an agent during the training process. Character $\mathbf{c}$ is a vector of character components $[c_1, \cdots, c_K]$. Each character component $c_k$ randomly sampled from pre-defined ranges, *e.g.*, $[0, 2.5]$. Sampled character is bounded fitting to significant figures, in our experiments we consider one significant figure.

---

> ### Author Response · Authors · 2024-08-07
>
> - **Standard deviation for main results and additional experiments on SMAC**
>
> We apologize for the inconvenience. We wanted to report it in the main body, but due to the page limit, we included it in the appendix of the original manuscript. Our appendix includes Tables with the standard deviation as follows.
> |Character diversity|n=1|n=2|n=3|n=4|n=5|
> |-|-|-|-|-|-|
> |Proposed|**2899** $\pm$ 217|**3047** $\pm$ 162|**2976** $\pm$ 196|**2948** $\pm$ 91|**3051** $\pm$ 109|
> |FCE-EFT|**2899** $\pm$ 217|2784 $\pm$ 161|2646 $\pm$ 196|2566 $\pm$ 103|2629 $\pm$ 125|
> |MADDPG|2763 $\pm$ 126|**3006** $\pm$ 103|2800 $\pm$ 106|**2933** $\pm$ 98|2856 $\pm$ 121|
> |MAPPO|2753 $\pm$ 206|2862 $\pm$ 201|2597 $\pm$ 144|2529 $\pm$ 131|2763 $\pm$ 190|
> |QMIX|2199 $\pm$ 56|2310 $\pm$ 39|2288 $\pm$ 118|2118 $\pm$ 82|1861 $\pm$ 132|
> |Dreamer|**2911** $\pm$ 312|2813 $\pm$ 283|2733 $\pm$ 351|2631 $\pm$ 521|2701 $\pm$ 433|
> |MBPO|2089 $\pm$ 804|1964 $\pm$ 753|1523 $\pm$ 948|1893 $\pm$ 792|1633 $\pm$ 821|
>
> |Algorithm|MAPPO|MADDPG|QMIX|Proposed|
> |-|-|-|-|-|
> |Spread|-149.29 $\pm$ 0.94|-157.10 $\pm$ 2.30|-154.70 $\pm$ 4.90|**-149.12** $\pm$ 1.38|
> |Adversary|9.61 $\pm$ 0.07|7.80 $\pm$ 1.43|8.11 $\pm$ 0.37|**10.01** $\pm$ 0.33|
> |Tag|13.78 $\pm$ 4.40|6.65 $\pm$ 3.90|**15.00** $\pm$ 2.73|14.57 $\pm$ 2.95|
>
> These tables show that **the standard deviation of EFTM is similar to that of other methods**. The model-based solution has the highest variance due to the uncertainty of other agents. Overall, EFTM achieves the best performance with a mid-level variance compared to all other baselines.
>
> Additionally, we have run additional experiments on SMAC [1], which is widely used for evaluating the MARL algorithm, to address the reviewer's concern. We report the performance ($4$ seeds) with MARL baselines, as follows.
> |SMAC Task|EFTM|MAPPO|MADDPG|QMIX|
> |-|-|-|-|-|
> | 2s3z|98.8 $\pm$ 2.3| **100** $\pm$ 1.5|90.3 $\pm$ 5.3|95.3 $\pm$ 2.5|
> | 3s5z vs 3s6z |**84.3** $\pm$ 9.1|63.3 $\pm$ 19.2|18.9 $\pm$ 4.8|82.8 $\pm$ 5.3|
> | 6h vs 8z|**93.8** $\pm$ 6.7|85.9 $\pm$ 30.9|68.0 $\pm$ 34.7|9.4 $\pm$ 2.0|
> | Total|**276.9**|249.2|177.2|187.5|
>
> This result also demonstrates that EFTM still has surpassing or matching performance with previous solutions. It means that EFTM is capable of generalizing to solve widely-used MARL tasks, achieving the best total scores. Notably, we set a simple setup for the SMAC and MPE environments that is, we follow a vanilla setup with a single character diversity $n=1$. **While fully leveraging the advantages of EFTM in these environments can be challenging, EFTM is nonetheless capable of delivering competitive performance in such settings.**
>
> ---
> Once again, we deeply appreciate the insightful comments and suggestions. We hope our clarification and additional empirical studies could address the concerns raised by the reviewer. Should there be any leftover questions, please let us know and we will make every effort to address them during the subsequent discussion period.

---

> > ### Comment · Reviewer_pTth · 2024-08-09
> > **Great!**
> >
> > Thanks for your follow-up work on this! The clarifications and new experiments greatly alleviate my concerns. In line with reviewer Cur7's thinking, I am upgrading my score to a 6.

---

> > > ### Author Response · Authors · 2024-08-10
> > >
> > > We sincerely appreciate the insights you’ve shared for this work and are grateful for your consideration in raising the score.

---

### Official Review · Reviewer_RgnJ · 2024-07-15

**Soundness:** 2
**Presentation:** 3
**Contribution:** 2
**Rating:** 6
**Confidence:** 4

**Summary:**

The paper introduces an Episodic Future Thinking (EFT) mechanism for reinforcement learning (RL) agents, inspired by cognitive processes observed in animals, to enhance social decision-making in multi-agent systems with diverse agent characteristics. The EFT mechanism uses a multi-character policy to infer the behavioral preferences of other agents, predicts their actions, and simulates potential future scenarios to select optimal actions. The authors evaluate the EFT mechanism in a multi-agent autonomous driving scenario and demonstrate that it leads to higher rewards and is robust across societies with varying levels of character diversity.

**Strengths:**

+ The paper introduces an episodic future thinking (EFT) mechanism for RL agents, borrowing from cognitive processes observed in animals, representing an interesting application of biological insights to enhance AI decision-making processes.

+ The multi-character policy handles both continuous and discrete action spaces, expanding the applicability of RL methods to more complex scenarios.

+ The paper demonstrates the effectiveness of the EFT mechanism in a multi-agent autonomous driving scenario. The authors examine the robustness of the EFT mechanism across different levels of character diversity, showing its resilience in various social compositions.

**Weaknesses:**

- The paper primarily focuses on an autonomous driving scenario. Demonstrating the EFT mechanism's effectiveness across a broader range of multi-agent scenarios, e.g., SMAC and VirtualHome, could strengthen the argument for its generalizability. While the paper mentions the mechanism's effectiveness across different levels of character diversity, a detailed scalability analysis in terms of the number of agents and interactions with human or heterogeneous agents could provide further confidence in the approach.
- The results in Table. 2 and .3 only report the average performance. It is necessary to report the standard deviation to make the results more confident, as the environments are highly dynamic and varying uncertainty.
- Lacking baselines. There are some works that incorporate ToM or opponent modeling with MARL[1,2]. It is necessary to compare those methods, e.g., estimate the current observation or hidden state of others instead of the next observation, to demonstrate the advantages of the proposed methods.


Ref:
[1] Agent modeling under partial observability for deep reinforcement learning. NeurIPS 2021
[2] ToM2C: Target-oriented Multi-agent Communication and Cooperation with Theory of Mind, ICLR 2022

**Questions:**

Q1: Do you know the concept of **role**  introduced in previous MARL works? is there any difference between the introduced character and role?

Q2: Can you validate the generalization of the agents by training them at a specific level and transferring them to other levels with unseen characters?

**Limitations:**

The authors discussed the limitations in the discussion.

---

> ### Author Rebuttal · Authors · 2024-08-07
>
> We appreciate the reviewer's detailed feedback and valuable suggestions for enhancing our work. In response, we describe how we have revised the paper to address the reviewer's concerns and questions.
>
> - **Additional experiment results to prove generalizability**
>
> To demonstrate the efficiency of the proposed solution, we would like to **report additional experimental outcomes on the SMAC (StarCraft multi-agent challenge)** [1], which is widely used in the multi-agent RL domain and recommended by the reviewer. The additional result ($4$ seeds) is as below.
> |SMAC Task|EFTM|MAPPO|MADDPG|QMIX|
> |-|-|-|-|-|
> | 2s3z|98.8 $\pm$ 2.3| **100** $\pm$ 1.5|90.3 $\pm$ 5.3|95.3 $\pm$ 2.5|
> | 3s5z vs 3s6z |**84.3** $\pm$ 9.1|63.3 $\pm$ 19.2|18.9 $\pm$ 4.8|82.8 $\pm$ 5.3|
> | 6h vs 8z|**93.8** $\pm$ 6.7|85.9 $\pm$ 30.9|68.0 $\pm$ 34.7|9.4 $\pm$ 2.0|
> | Total|**276.9**|249.2|177.2|187.5|
>
> The table above shows that EFTM still has surpassing or matching performance with previous solutions. **It means that EFTM is capable of generalizing to solve widely-used MARL tasks, achieving the best total scores.** More precisely, we set a simple setup for the SMAC environment, akin to the MPE task, that is we follow a vanilla setup with a single character diversity $n=1$. The reported scores of MAPPO and QMIX are based on benchmark performance [2].
>
> We also checked VirtualHome [3] as the reviewer recommended, but it seemed inappropriate for us due to the requirement of the language model, so we decided not to use this task. Although this response did not cover VirtualHome [3], we believe that our results in three tasks, e.g., autonomous driving, MPE, and SMAC, can alleviate the concern of the reviewer. Thank you for your suggestion, which helps us to prove the generalization of our method.
>
> ---
> - **Experiments of additional baselines - theory of mind and agent modeling**
>
> We thank the reviewer for a valuable suggestion about comparison baselines. As per the reviewer’s suggestion, **we have run experiments on additional baselines, including ToM2C [4] and opponent modeling [5], as follows.**
> |character diversity|n=1|n=2|n=3|n=4|n=5|
> |-|-|-|-|-|-|
> |EFTM|2899 $\pm$ 217|**3047** $\pm$ 162|**2976** $\pm$ 196|**2948** $\pm$ 91|**3051** $\pm$ 109|
> |ToMC2 [4]|**3016** $\pm$ 109|2812 $\pm$ 273|2683 $\pm$ 309|2691 $\pm$ 458|2511 $\pm$ 397|
> |Opponent Modeling [5]|1913 $\pm$ 330|1792 $\pm$ 410|1771 $\pm$ 367|1683 $\pm$ 381|1733 $\pm$ 429|
>
> **This result demonstrates the effectiveness and adaptability of our approach, achieving higher rewards when character diversity exists.** ToMC2 achieves the best score in the $n=1$ scenario, but its performance decreases as the diversity level increases; opponent modeling fails at all diversity levels. On the other hand, the proposed solution is robust to changes in the surrounding agents and maintains high performance across diversity levels.
>
> We conjecture why two baselines fail in this setup, as follows. **ToMC2 requires retraining or adjusting the ToM module as surrounding agents change**. The ToM module is tailored to other agents for the prediction of information (*e.g.*, goals, observations, and actions). Next, **opponent modeling also necessitates a new opponent modeling process for each test environment**. In addition, prior works on opponent modeling rarely involve more than four players. In contrast, our selective tasks consider $20$ surrounding agents and their policies can be subject to change.
>
> ---
> - **Validate the generalizability of the EFT agent**
> We appreciate the reviewer's valuable feedback about the generalizability of the multi-character policy over unseen characters. As the reviewer pointed out, validating the generalizability of the proposed solution is important for realistic tasks. To verify this, we have run additional experiments under the following two cases:
>
> 1) Case 1: Train multi-character policy by experiencing character range $[0.0, 0.6]$ and $[0.8, 1.0]$, then test the accuracy of unseen character inference over unseen characters $\{0.65, 0.7, 0.75\}$.
> 2) Case 2: Train multi-character policy by experiencing character range $[0.2, 0.8]$, then test the accuracy of unseen character inference over unseen characters $\{0.0, 0.1, 0.9, 1.0\}$.
>
> Below are additional results in terms of character inference ($20$ inference trials).
> |True character|0.65 (case1)|0.7 (case1)|0.75 (case1)|0.0 (case2)|0.1 (case2)|0.9 (case2)|1.0 (case2)|
> |-|-|-|-|-|-|-|-|
> |Inferred character|0.61 $\pm$ 0.09|0.67 $\pm$ 0.15|0.76 $\pm$ 0.08|0.12 $\pm$ 0.21|0.13 $\pm$ 0.04|0.85 $\pm$ 0.13|0.82 $\pm$ 0.28|
>
> For case 1, the inferred characters are reasonably close to their true values. This indicates that the policy could have partial generalizability through interpolation, even for values within the gap not explicitly covered by the training ranges. For case 2, as the actual character gets farther away from the experienced value, it loses generalization performance, increasing the standard deviation and gap between the inferred and true one.
>
> To avoid encountering unseen values as much as possible, we should set a realistic character range and deeply consider the sampling method during the learning process. In our case, we used uniform random sampling so that diverse characters could be experienced evenly within the predefined character range. Additionally, we believe that some few-shot learning and adaptation methods can alleviate these problems.

---

> > ### Comment · Reviewer_RgnJ · 2024-08-11
> > **Thanks for you response**
> >
> > My main concern about the generalization has been addressed in the response. I tend to maintain my rate, as I think further clarification on the details of the experiments is required.
> > - Can you explain the implementation details of the ToM2C and Opponent model in your experiment?
> > - Do you have any idea on building a more general computation model that combines the role and character jointly in the agent?
> > - The Virtualhome environment does not need language model at all. There are also some other simulators close to VirtualHome, such as 3DWorld. If you can not extend your model on such 3D environments, can you explain the reasons or how to extend the current version for these 3D environments?

---

> > > ### Author Response · Authors · 2024-08-12
> > >
> > > Thank you for your active response! To ease any remaining concerns, we leave our opinions on additional questions below.
> > >
> > > ---
> > > **Experimental details**
> > >
> > > Thank you for this comment. Our implementation follows the official Git repositories from ToM2C [4] and opponent modeling [5] (In accordance with NeurIPS 2024 policy, we cannot upload hyperlinks in OpenReview). Given that we consider the POMDP setup, it is important to set how many other agents that an agent has access to. For ToM2C, we consider full access in accordance with the paper: they reported that full access has better performance than partial access. On the other hand, for opponent modeling, we consider six surrounding vehicles, not entire agents. That is because the reference paper aims to model the other agents in local information. Finally, training and validation setups are the same as other baselines.
> > >
> > > ---
> > > **Role and Character**
> > >
> > > Thank you for this constructive comment regarding the future direction of our community. A promising approach for combining the concepts of role and character would be to use a hierarchical structure. Each agent within a cooperative team first defines its role or subtask. The agent could then decide on the most effective strategy to achieve its subgoal, taking into account the characters and behaviors of other agents. We genuinely believe that this approach could be valuable in various studies, e.g., multi-agent planning tasks, as it emphasizes setting broad objectives first and then making detailed decisions based on interactions within the multi-agent system.
> > >
> > > ---
> > > **3D Environments**
> > >
> > > We apologize for our mis-clarification regarding the language models in VirtualHome. The authors of VirtualHome [3] reported they consider video with text, so we have a misunderstanding about the need for a language model. Thank you for your correction, and it may not strictly require a language model.
> > >
> > > We believe our concept could still be relevant to the testbeds you suggested. Since these environments are based on images or video, they would require more advanced forward prediction and representation networks to manage the complexities of 3D data. Specifically, VirtualHome operates in a 2D or 3D observation space, requiring at least 64 x 64 x 3 features as input. In contrast, SMAC and MPE tasks use a 1D observation space with about 100-200 and 10-20 features, respectively. By implementing an appropriate module for handling 3D data, our model could be extended to function in these more demanding 3D environments.
> > >
> > > We deeply acknowledge the value and importance of the reviewer’s request, so we would like to explore additional results in various domains. Regrettably, our group has limited GPU resources, unlike tech companies, making it challenging to get results for more computationally intensive tasks. At the same time, while applying our work to 3D environments is relevant, we believe that it is not the most critical aspect of our work. Our main focus is to develop a social decision-making process in a heterogeneous society where multiple characteristics coexist. We claim that the value of our method has been fully demonstrated in testbeds such as autonomous driving tasks, MPE, and SMAC. Sorry again that we could not include VirtualHome results, and we would greatly appreciate your understanding of our computational resource limitation.
> > >
> > > If you have any other questions or comments that could raise your score, we would be happy to continue the discussion, given the time!
> > >
> > > [3] P. Xavier et al., Virtualhome: Simulating household activities via programs. CVPR 2018.
> > >
> > > [4] Y. Wang et al., ToM2C: Target-oriented multi-agent communication and cooperation with theory of mind. ICLR 2022.
> > >
> > > [5] P. Georgios et al., Agent modelling under partial observability for deep reinforcement learning. NeurIPS 2021.

---

> > > > ### Comment · Reviewer_RgnJ · 2024-08-13
> > > >
> > > > Thanks for your detailed response. I have upgraded my rating to 6.

---

> > > > > ### Author Response · Authors · 2024-08-13
> > > > >
> > > > > We deeply appreciate the valuable discussion for this work and are thankful for raising the score!

---

> ### Author Response · Authors · 2024-08-07
>
> - **Difference between role and character**
>
> We appreciate the reviewer for bringing out insightful discussion. A 'role' in a multi-agent system represents a responsibility or function for achieving the objective of a cooperative team [6]. Roles can be interpreted as subtasks for each agent. A 'character' refers to the specific behavioral strategies an agent employs to perform its assigned role.
>
> To illustrate, suppose a cooperative multi-agent task, including two different roles necessary to achieve the team’s goal, and two agents, being assigned a specific role. The agent aims to solve its subtask which can be solved using different strategies. The character endows a behavioral preference to the agent.
>
> These two concepts are considered and debated significantly in the MARL domain. While 'role' has been the focus of several prior works, 'character' concept remains relatively overlooked. **We sincerely emphasize that it is essential to consider a task with multiple agents with diverse characteristics in the MARL community.** We believe that our work can serve the beginning, and the broader impact on the community will be meaningful. Taking this into account, we will include extensive related works about 'role' and 'character' in the appendix of the final version.
>
> ---
> - **Standard deviation for main results**
>
> We apologize for the inconvenience. We wanted to report it in the main body, but due to the page limit, we included it in the appendix of the original manuscript. Our appendix includes Tables with the standard deviation as follows.
> |Character diversity|n=1|n=2|n=3|n=4|n=5|
> |-|-|-|-|-|-|
> |Proposed|**2899** $\pm$ 217|**3047** $\pm$ 162|**2976** $\pm$ 196|**2948** $\pm$ 91|**3051** $\pm$ 109|
> |FCE-EFT|**2899** $\pm$ 217|2784 $\pm$ 161|2646 $\pm$ 196|2566 $\pm$ 103|2629 $\pm$ 125|
> |MADDPG|2763 $\pm$ 126|**3006** $\pm$ 103|2800 $\pm$ 106|**2933** $\pm$ 98|2856 $\pm$ 121|
> |MAPPO|2753 $\pm$ 206|2862 $\pm$ 201|2597 $\pm$ 144|2529 $\pm$ 131|2763 $\pm$ 190|
> |QMIX|2199 $\pm$ 56|2310 $\pm$ 39|2288 $\pm$ 118|2118 $\pm$ 82|1861 $\pm$ 132|
> |Dreamer|**2911** $\pm$ 312|2813 $\pm$ 283|2733 $\pm$ 351|2631 $\pm$ 521|2701 $\pm$ 433|
> |MBPO|2089 $\pm$ 804|1964 $\pm$ 753|1523 $\pm$ 948|1893 $\pm$ 792|1633 $\pm$ 821|
>
> |Algorithm|MAPPO|MADDPG|QMIX|Proposed|
> |-|-|-|-|-|
> |Spread|-149.29 $\pm$ 0.94|-157.10 $\pm$ 2.30|-154.70 $\pm$ 4.90|**-149.12** $\pm$ 1.38|
> |Adversary|9.61 $\pm$ 0.07|7.80 $\pm$ 1.43|8.11 $\pm$ 0.37|**10.01** $\pm$ 0.33|
> |Tag|13.78 $\pm$ 4.40|6.65 $\pm$ 3.90|**15.00** $\pm$ 2.73|14.57 $\pm$ 2.95|
>
> These tables show that **the standard deviation of EFTM is similar to that of other methods.** The model-based solution has the highest variance due to the uncertainty of other agents. Overall, EFTM achieves the best performance with a mid-level variance compared to all other baselines.
>
> ---
> Once again, we deeply appreciate the insightful comments and suggestions. We hope our clarification and additional empirical studies could address the concerns raised by the reviewer. Should there be any leftover questions, please let us know and we will make every effort to address them during the subsequent discussion period.

---

### Author Rebuttal · Authors · 2024-08-07

We express our gratitude to all five reviewers for their insightful feedback. We are pleased to present the updates we have made in response to valuable suggestions, as detailed below.

- We compared the performance with **two additional baselines** on the research about opponent modeling and theory of mind (Reviewer RgnJ). This result confirmed the effectiveness and adaptability of our approach, outperforming additional baselines.
- We evaluated EFTM and baselines on **SMAC (StarCraft multi-agent challenge)**, which demonstrates the generalizability of EFTM for the MARL domain (Reviewer RgnJ). This result demonstrated that the proposed solution still has comparable performance in a widely used MARL setup.
- We added **time complexity analyses** in the inference phase in terms of big $\mathcal O$ analysis and wall-clock time measure (Reviewer Cur7 and ukHS).

Next, we apologize for not including standard deviations for performance evaluations in the main body of the paper. Due to the page limit, we included **the results with standard deviations in Appendix J2 and K2 of the original manuscript** (Reviewer RgnJ, pTth, and cur7).

Below are the references we use in this response.

[1] S. Mikayel et al., The Starcraft multi-agent challenge. AAMAS 2019.

[2] Y. Chao et al., The surprising effectiveness of PPO in cooperative multi-agent games. NeurIPS 2022.

[3] P. Xavier et al., Virtualhome: Simulating household activities via programs. CVPR 2018.

[4] Y. Wang et al., ToM2C: Target-oriented multi-agent communication and cooperation with theory of mind. ICLR 2022.

[5] P. Georgios et al., Agent modelling under partial observability for deep reinforcement learning. NeurIPS 2021.

[6] Z. Xianghua et al., Effective and stable role-based multi-agent collaboration by structural information principles. AAAI 2023.

[7] G. Papoudakis et al., Benchmarking multi-agent deep reinforcement learning algorithms in cooperative tasks. NeurIPS 2020.

[8] H. Danijar et al., Dream to control: Learning behaviors by latent imagination. ICLR 2020.

[9] H. Danijar et al., Mastering Atari with discrete world models. ICLR 2021.

[10] H. Danijar et al., Mastering diverse domains through world models. arXiv preprint arXiv:2301.04104 (2023).

[11] M. Volodymyr et al., Human-level control through deep reinforcement learning. Nature 2015.

[12] T. Yuval et al., Deepmind control suite. arXiv preprint arXiv:1801.00690 (2018).

[13] B. Charles et al., Deepmind lab. arXiv preprint arXiv:1612.03801 (2016).

[14] S. Satinder et al., Learning without state-estimation in partially observable Markovian decision processes. Machine Learning Proceedings 1994.

[15] G. Jim. How to: Define minimum SNR values for signal coverage. Viitattu 23 (2012).

---

### Decision · Program_Chairs · 2024-09-25

**Decision:**

Accept (poster)

**Comment:**

Reviewers unanimous agree on accepting the paper, with scores of (5,6,6,6,6). These scores were reached after the authors provided a detailed rebuttal with a large number of new experiments and results that directly addressed reviewers’ concerns. Reviewers engaged in a discussion with the authors and in several cases raised their scores.

**Summary of reviewer opinions**

Reviewers praised the following strengths:
- They believed the idea for the method is a good one, and several called it a “significant contribution” (ukHS,Cur7) . “A policy that can handle diverse agent characters is a significant contribution. This addresses a real challenge in multi-agent systems where agents may have different goals or behavioral traits.” (Cur7)
- Inspiration from real cognitive processes in mammals (RgnJ); “The cognitive motivation makes a lot of sense” (pTth)
- Thorough experiments. “multiple reasonable baselines” (Cur7). “The studies of 5.2 and 5.3 are welcome additions that help make sense of how the method works.” (pTth).  Experiments show robustness across different scenarios (RgnJ)
- Clearly written and structured (pTth, Cur7,ukHS)
- “handles both continuous and discrete action spaces” (RgnJ)

Reviewers complained about the following weaknesses:
- “The driver environment has been designed so that the proposed method has precisely the right inductive bias” (pTth) (essentially, the driving experiment is set up so that other drivers have “characters” and this method is the right solution for that scenario)
- Not enough environments (original main paper was autonomous driving and MPE) (RgnJ)
- Complaints about insufficient baselines (pTth, RgnJ) (e.g. lacking baselines that incorporate ToM or opponent modeling with MARL (RgnJ))
- Lack of scalability / computational complexity analysis (RgnJ, Cur7,ukHS)
- Need standard deviations / confidence intervals in the results (RgnJ, pTth, Cur7)

**Effect of rebuttal**

In the rebuttal, authors added experiments which directly addressed the reviewers’ concerns, including comparing to two new baselines, obtaining results on a new environment (SMAC), giving baselines more experience with diverse characters, increasing the proportion of EFT agents, and some new generalization experiments. They also added a big Oh complexity analysis, and measured the wall clock time. Finally, they pointed reviewers to additional results with standard deviations and 3 additional environments provided in the appendix. The rebuttal also had good explanations for why certain decisions were made (e.g. choosing Dreamer v1 as opposed to v2-3 as a baseline).

**Final recommendation**: Accept